# Stochastic Optimization Algorithms for Instrumental Variable Regression with Streaming Data

**Xuxing Chen**[♯]
Department of Mathematics
University of California, Davis
xuxchen@ucdavis.edu

**Abhishek Roy**[♯]
Halıcıoğlu Data Science Institute
University of California, San Diego
a2roy@ucsd.edu

**Yifan Hu**
College of Management, EPFL
Department of Computer Science, ETH Zurich
yifan.hu@epfl.ch

**Krishnakumar Balasubramanian**
Department of Statistics
University of California, Davis
kbala@ucdavis.edu

## Abstract

We develop and analyze algorithms for instrumental variable regression by viewing the problem as a conditional stochastic optimization problem. In the context of least-squares instrumental variable regression, our algorithms neither require matrix inversions nor mini-batches and provides a fully online approach for performing instrumental variable regression with streaming data. When the true model is linear, we derive rates of convergence in expectation, that are of order $\mathcal{O}(\log T/T)$ and $\mathcal{O}(1/T^{1-\iota})$ for any $\iota > 0$, respectively under the availability of two-sample and one-sample oracles, where $T$ is the number of iterations. Importantly, under the availability of the two-sample oracle, our procedure avoids explicitly modeling and estimating the relationship between the independent and the instrumental variables, demonstrating the benefit of the proposed approach over recent works based on reformulating the problem as minimax optimization problems. Numerical experiments are provided to corroborate the theoretical results.

## 1 Introduction

Instrumental variable analysis is widely used in fields like econometrics, health care [TM17], social science [Bol12], and online advertisement to estimate the causal effect of a random variable, $X$, on an outcome variable, $Y$, when an unobservable confounder influences both. By identifying an instrumental variable correlated with the variable $X$ but unrelated to the confounders, researchers can isolate the exogenous variation in $X$ and estimate a causal relationship between $X$ and $Y$. In the context of regression, Instrumental Variable Regression (IVaR) addresses endogeneity issues when an independent variable is correlated with the error term in the regression model, leveraging an instrument variable $Z$ such that $Y$ is independent of $X|Z$. In this paper, we focus on the following statistical model:

$$Y = g_{\theta^*}(X) + \epsilon_1 \quad \text{with} \quad X = h_{\gamma^*}(Z) + \epsilon_2 \tag{1}$$

where $X \in \mathbb{R}^{d_x}$ and $\epsilon_1$ are correlated and $\epsilon_2$ is a centered unobserved noise (independent of $Z \in \mathbb{R}^{d_z}$), leading to confounding in the model between $X$ and $Y \in \mathbb{R}$. Here $\epsilon_1$ and $\epsilon_2$ are dependent, and $\theta^*$ and $\gamma^*$ are true parameters for the respective function $g$ and $h$. Our goal is to design efficient algorithms that recovers $\theta^*$ from the data.

---

[♯]XC and AR contributed equally to this work.

38th Conference on Neural Information Processing Systems (NeurIPS 2024).

Traditionally, IVaR algorithms are based on two-stage estimation procedures, where we first regress $Z$ and $X$ to obtain an estimator $\widehat{X}$, and then regress $\widehat{X}$ and $Y$, with the essence that $\widehat{X}$ is independent of $Y$, and thus eliminating the aforementioned endogeneity of the unknown confounder. A vast literature has devoted to understanding the two-stage approaches [HH05, DFFR11, HLLBT17], with the parametric two-stage least-squares (2SLS) procedure being the most canonical one [AI95]. The main drawback of this approach is that the second-stage regression problem is affected by the estimation error from the regression problem corresponding to first stage. In fact, [AP09] call the first stage regression as "forbidden regression", due to the concerns in estimating a nuisance parameter.

Considering the squared loss function, [MMLR20] formulate the IVaR problem as a conditional stochastic optimization problem [HZCH20]:

$$\min_{g \in \mathcal{G}} F(g) := \mathbb{E}_Z \mathbb{E}_{Y|Z}[(Y - \mathbb{E}_{X|Z}[g(X)])^2]. \tag{2}$$

However, [MMLR20] did not solve problem (2) efficiently, and resort to reformulating (2) further as a minimax optimization problem. Indeed, they mention explicitly in their work that "*it remains cumbersome to solve* (2) *directly because of the inner expectation*". Then, they leverage the Fenchel conjugate of the squared loss, leading to a minimax optimization with maximization over a continuous functional space. Following [DHP$^+$17], [MMLR20] propose to use reproducing kernel Hilbert space (RKHS) to handle the maximization over continuous functional space. See also [LS18, BKS19, DLMS20, LCY$^+$20, BKM$^+$23] for similar minimax approaches. The issue with such an approach is that approximating the dual variable via maximization over continuous functional space inevitably introduces approximation error. Hence, although there is no explicit nuisance parameter estimation step like in the two-stage approach, there is an implicit one, which makes the minimax approach less appealing as an alternate to the two-stage procedures.

In this work, contrary to the claim made in [MMLR20] that problem (2) is cumbersome to solve, we design and analyze efficient streaming algorithms to directly solve the conditional stochastic optimization problem in (2). Direct application of methods from [HZCH20] for solving (2) is possible, yet their approach utilizes nested sampling, i.e., for each sample of $Z$, [HZCH20] generate a batch of samples of $X$ from $\mathbb{P}(X|Z)$, to reduce the bias in estimating the composition of non-linear loss function with conditional expectations. Thus their methods are not suitable for the streaming setting that we are interested in. Considering (2), we first parameterize the function class $\mathcal{G} := \{g(\theta; X) \mid \theta \in \mathbb{R}^{d_\theta}\}$. Now, defining $F(g) := F(\theta)$, we observe that the gradient $\nabla F(\theta)$ admits the following form

$$\nabla F(\theta) = \mathbb{E}_Z[(\mathbb{E}_{X|Z}[g(\theta; X)] - \mathbb{E}_{Y|Z}[Y])\nabla_\theta \mathbb{E}_{X|Z}[g(\theta; X)]], \tag{3}$$

which implies that one does not need the nested sampling technique to reduce the bias. However, the presence of product of two conditional expectations $\mathbb{E}_{X|Z}[g(\theta; X)]$ still causes significant challenges in developing stochastic estimators of the above gradient in the streaming setting. In this work, we overcome this challenge and develop two algorithms that are applicable to the streaming data setting avoiding the need for generating batches of samples of $X$ from $\mathbb{P}(Z|X)$.

**Contributions.** We make the following contributions in this work.

- **Two-sample oracles:** Our first algorithm leverages the observation that if we have access to a two-sample oracle that outputs *two* samples $X$ and $X'$ that are independent conditioned on the instrument $Z$, we can immediately construct an unbiased stochastic gradient estimator of the gradient in (3). Based on this crucial observation, we propose the *Two-Sample One-stage Stochastic Gradient IVaR* (TOSG-IVaR) method (Algorithm 1) that avoids explicitly having to estimate or model the relationship between $Z$ and $X$ thereby overcoming the "forbidden regression" problem.. Under standard statistical model assumptions, for the case when $g$ is a linear model, we establish rates of convergence of order $\mathcal{O}(\log T/T)$ for the proposed method, where $T$ is the overall number of iterations; see Theorem 1.

- **One-sample oracles:** In the case when we do not have the aforementioned two-sample oracle, we estimate the stochastic gradient in (3) by using the streaming data to estimate one of the conditional expectations, and the corresponding prediction to estimate the other, resulting in the *One-Sample Two-stage Stochastic Gradient IVaR* (OTSG-IVaR) method (Algorithm 2). Assuming further that the $X$ depends linearly on the instrument $Z$, we establish a rate of convergence of order $\mathcal{O}(1/T^{1-\iota})$, for any $\iota > 0$; see Theorem 2.

## 1.1 Literature Review

**IVaR analysis.** Instrumental variable analysis has a long history, starting from the early works by [Wri28] and [Rei45]. Several works considered the aforementioned two-stage procedure for IVaR; a summary could be found in the work by [AP09]. Nonparametric approaches based on wavelets, splines, reproducing kernels and deep neural networks could be found, for example, in the works by [HLLBT17, SSG19, BKS19, MMLR20, MZG+21, XCS+21, ZGG+22, PSF24]. Another popular approach for IVaR is via Generalized Method of Moments (GMM); see, for example, [CP12, BKS19, DLMS20] for an overview. Such approaches essentially reformulate the problem as a minimax problem and hence suffer from the aforementioned "forbidden regression" problem.

**Identifiability conditions for IVaR.** Several works in the literature have also focused on establishing the identifiability conditions for IVaR in the parametric and the nonparametric setting. Regardless of the procedure used, they are invariably based on certain source conditions motivated by the inverse problems literature (see, for example, [CFR07, CR11, BKM+23]) or the related problem of completeness conditions, which posits that the conditional expectation operator is one-to-one [BF17, LCY+20]. Semi-parametric identifiability is also considered recently in the work of [CPS+23]. Our focus in this work is not focused on the identifiability; for the formulation (2) that we consider, [MMLR20] provide necessary conditions for identifiability that we adopt.

**Stochastic optimization with nested expectations.** Recently, much attention in the stochastic optimization literature has focused on optimizing a nested composition of $T$ expectation functions. Sample average approximation algorithms in this context are considered in the works of [EN13] and [HCH20]. Optimal iterative stochastic optimization algorithms for the case of $T = 2$ were by derived by [GRW20]. For the general $T \geq 1$ case, [WFL17] provided sub-optimal rates, whereas [BGN22] derived optimal rates; see also [ZX21] and [CSY21] for related works under stronger assumptions, and [Rus21] for similar asymptotic results. While the above works required certain independence assumptions regarding the randomness across the different compositions, [HZCH20, HWX+24] studied the case of $T = 2$ where the the randomness are generically dependent. They termed this problem setting as conditional stochastic optimization, which is the framework that the IVaR problem in (2) falls in. Compared to prior works, for e.g., [GRW20] and [BGN22], in order to handle the dependency between the levels, [HZCH20] require mini-batches in each iteration, making their algorithm not immediately applicable to the purely streaming setting. In this work, we show that despite the problem (2) being a conditional stochastic optimization problem, mini-batches are not required due the additional favorable quadratic structure available in IVaR.

**Streaming IVaR.** [VSH+16, DVB24] analyzed streaming versions of 2SLS in the online[1] and adversarial settings. Focusing on linear models, [VSH+16] provide preliminary asymptotic analysis assuming access to efficient *no-regret learners*, while [DVB24] provide regret bounds under the strong assumption that the instrument is almost surely bounded. Furthermore, our algorithms have significantly improved per-iteration and memory complexity compared to [DVB24]; see Sections A and B for details. [CLL+23] developed stochastic optimization algorithms for the GMM formulation and provide asymptotic analysis. Their algorithm requires access to an offline dataset for initialization and is hence not fully online. The above works (i) do not focus on avoiding the forbidden regression problem and (ii) do not view IVaR via the *conditional stochastic optimization* lens, like we do.

## 2 Two-sample One-stage Stochastic Gradient Method for IVaR

Recall that our goal is to solve the objective function given in (2). By [MMLR20, Theorem 4], the optimal solution of (2) gives the true underlying causal relationship under the following assumption.

**Assumption 2.1.** *(Identifiability Assumption)*

- *The conditional distribution $\mathbb{P}_{Z|X}$ is continuous in $Z$ for any value of $X$.*

---

[1]Their notion of online is from the literature on *online learning* [SS+12].

---

**Algorithm 1** Two-sample One-stage Stochastic Gradient-IVaR (`TOSG-IVaR`)

---

**Input:** ♯ of iterations $T$, stepsizes $\{\alpha_t\}_{t=1}^T$, initial iterate $\theta_1$.
1: **for** $t = 1$ to $T$ **do**
2:     Sample $Z_t$, sample independently $X_t$ and $X_t'$ from $\mathbb{P}_{X|Z_t}$, and sample $Y_t$ from $\mathbb{P}_{Y|X_t}$.
3:     Update $\theta_t$
$$\theta_{t+1} = \theta_t - \alpha_{t+1}(g(\theta_t; X_t) - Y_t)\nabla_\theta g(\theta_t; X_t').$$
4: **end for**
**Output:** $\theta_T$.

---

- *The function class $\mathcal{G} := \{g(\theta; X) \mid \theta \in \mathbb{R}^{d_\theta}\}$ is correctly specified, i.e., it includes the true underlying relationship between $X$ and $Y$.*

Notice that both assumptions are standard in the IVaR literature [NP03, CP12, MMLR20], and makes the objective in (2) is the meaningful for IVaR. However, [MMLR20] resort to reformulating the objective function in (2) as a minimax optimization problem as described in Section 1. While their original motivation was to avoid two-state estimation procedure and avoid the "forbidden regression", their minimax reformulation ends up having to solve a complicated approximation of the original objective resulting in having to characterize the approximation error which is non-trivial.

**Algorithm and Analysis.** Our aim in this work is to directly solve the original problem in (2), leveraging the structure provided by the quadratic loss. Given the gradient formulation in (3), a natural way to build unbiased gradient estimator is to generate $X$ and $X'$, two independent samples of $X$ from the conditional distributions $\mathbb{P}_{X|Z}$, for a given realization of $Z$ and generate one sample of $Y$ from the conditional distribution $\mathbb{P}_{Y|X}$. Then, an unbiased gradient estimator is

$$v(\theta) = (g(\theta; X) - Y)\nabla_\theta g(\theta; X'). \tag{4}$$

This could be plugged into the standard stochasic gradient descent algorithm, which give us the Two-sample Stochastic Gradient Method for IVR (TSG-IVaR) method illustrated in Algorithm 1. In particular, the algorithm never requires estimating (or modeling) the relationship between $X$ and $Z$ as needed in the two-stage procedure [AP09] and the minimax formulation based procedures [MMLR20, LS18, BKS19, DLMS20, LCY+20, BKM+23]. Furthermore, this viewpoint not only provides a novel algorithm for performing IV regression, but also provides a novel data collection mechanism for the practical implementation of IVaR. In addition, such a two-sample gradient method is not very restrictive when the instrumental variable $Z$ takes value in a discrete set. In this case, to implement the two-sample oracle, it is enough simply pick two sets of samples $(X, Y, Z)$ and $(X', Y', Z)$ for which $Z$ has repeated observations (which is possible when $Z$ is a discrete random variable) from a pre-collected dataset.

**Assumption 2.2.** *The tuple $(Z_t, X_t, X_t', Y_t)$ is independent and identically distributed, across $t$.*

To demonstrate the convergence rate of Algorithm 1, we first consider the case when $g$ is a linear function, i.e., $g(\theta; X) = X^\top \theta$. We make the following assumptions.

**Assumption 2.3.** *Suppose there exists $\mu > 0$ such that $\mathbb{E}_Z\left[\mathbb{E}_{X|Z}[X] \cdot \mathbb{E}_{X|Z}[X]^\top\right] \succeq \mu I$.*

**Assumption 2.4.** *Let $(\vartheta_1, \vartheta_2, \vartheta_3, \vartheta_4) \in \mathbb{R}_+^4$. For any $Z$, $X'$ and $X$ i.i.d. generated from $\mathbb{P}_{Z|X}$, and $Y$ generated from $\mathbb{P}_{Y|X}$. There exists constants $C_x, C_y, C_{xx}, C_{yx} > 0$ such that*

$$\mathbb{E}\left[\|X'X^\top - \mathbb{E}_{X|Z}[X]\mathbb{E}_{X|Z}[X]^\top\|^2\right] \leq C_x d_x^{\vartheta_1}, \tag{5}$$

$$\mathbb{E}\left[\|YX' - \mathbb{E}_{Y|Z}[Y]\mathbb{E}_{X|Z}[X]\|^2\right] \leq C_y d_x^{\vartheta_2}, \tag{6}$$

$$\mathbb{E}\left[\|\mathbb{E}_{X|Z}[X] \cdot \mathbb{E}_{X|Z}[X]^\top - \mathbb{E}_Z\left[\mathbb{E}_{X|Z}[X] \cdot \mathbb{E}_{X|Z}[X]^\top\right]\|^2\right] \leq C_{xx} d_z^{\vartheta_3}, \tag{7}$$

$$\mathbb{E}\left[\|\mathbb{E}_{Y|Z}[Y] \cdot \mathbb{E}_{X|Z}[X] - \mathbb{E}_Z\left[\mathbb{E}_{Y|Z}[Y] \cdot \mathbb{E}_{X|Z}[X]\right]\|^2\right] \leq C_{yx} d_z^{\vartheta_4}, \tag{8}$$

*where $\|\cdot\|$ denotes the Euclidean norm and operator norm for a vector and matrix respectively.*

The above assumptions are mild moment assumptions required on the involved random variables. The following result demonstrates that Assumptions 2.3 and 2.4 are naturally satisfied even under non-linear modeling assumption on (1). We defer its proof to Section D.3.

**Lemma 1.** *Suppose there exist* $\theta_* \in \mathbb{R}^{d_x}$, $\gamma_* \in \mathbb{R}^{d_z \times d_x}$, *a non-linear map* $\phi : \mathbb{R}^{d_x} \to \mathbb{R}^{d_x}$, *and a positive semi-definite matrix* $\Sigma \in \mathbb{R}^{d_z \times d_z}$ *such that*

$$
\mathbb{E}_Z\left[\phi(\gamma_*^\top Z) \cdot \phi(\gamma_*^\top Z)^\top\right] \succeq \mu I, \ \mathbb{E}[\|\phi(\gamma_*^\top Z)\|^2] = \mathcal{O}(d_x),
$$

$$
Z \sim \mathcal{N}(0, \Sigma), \ X = \phi(\gamma_*^\top Z) + \epsilon_2, \ Y = \theta_*^\top X + \epsilon_1, \ \epsilon_2 \sim \mathcal{N}(0, \sigma_{\epsilon_2}^2 I_{d_x}), \ \epsilon_1 \sim \mathcal{N}(0, \sigma_{\epsilon_1}^2), \quad (9)
$$

*where* $\epsilon_1, \epsilon_2$ *are independent of* $Z$ *and*

$$
\mathbb{E}\left[\epsilon_1^2 \|\epsilon_2\|^2\right] \leq \sigma_{\epsilon_1, \epsilon_2}^2 d_x, \ \mathbb{E}\left[\|\phi(\gamma_*^\top Z) \cdot \phi(\gamma_*^\top Z)^\top - \mathbb{E}[\phi(\gamma_*^\top Z) \cdot \phi(\gamma_*^\top Z)^\top]\|^2\right] \leq C d_z. \quad (10)
$$

*Then Assumptions 2.3 and 2.4 hold with* $\vartheta_1 = \vartheta_2 = 2$ *and* $\vartheta_3 = \vartheta_4 = 1$. *If* $\phi$ *is an identity map, then the conditions involving* $\phi$ *become* $\gamma_*^\top \Sigma \gamma_* \succeq \mu I$, $\text{tr}(\gamma_*^\top \Sigma \gamma_*) = \mathcal{O}(d_x)$, $\mathbb{E}\left[\|ZZ^\top - \Sigma\|^2\right] \leq C d_z$.

The above assumption is standard in the stochastic approximation, statistics and econometrics literature. It could be further relaxed to Markovian-type dependency assumptions, following techniques in the works of [DAJJ12, SSY18, Eve23, RBG22]; we leave a detailed examination of the Markovian streaming setup as future work. Under the above assumptions, we have the following result demonstrating the last-iterate global convergence of Algorithm 1.

**Theorem 1.** *Suppose Assumptions 2.3, 2.4, and 2.2 hold. In Algorithm 1, defining* $\sigma_1^2 := 2C_x d_x^{\vartheta_1} + 2C_{xx} d_z^{\vartheta_3}$ *and* $\sigma_2^2 := C_y d_x^{\vartheta_2} + C_{yx} d_z^{\vartheta_4}$, *set* $\alpha_t \equiv \alpha = \frac{\log T}{\mu T} \leq \frac{\mu}{\mu^2 + 3\sigma_1^2}$. *Then, we have*

$$
\mathbb{E}\left[\|\theta_T - \theta_*\|^2\right] \leq \frac{\mathbb{E}\left[\|\theta_0 - \theta_*\|^2\right]}{T} + \frac{3\|\theta_*\|^2(\sigma_1^2 + \sigma_2^2)\log T}{\mu^2 T}.
$$

**Proof techniques.** In the analysis of Theorem 1, the following decomposition (see (18) for the derivation) plays a crucial role:

$$
\theta_{t+1} - \theta_* = A_t + \alpha_{t+1} B_t,
$$

$$
A_t = \theta_t - \alpha_{t+1}\mathbb{E}_Z\left[\mathbb{E}_{X|Z}[X] \cdot \mathbb{E}_{X|Z}[X]^\top\right]\theta_t + \alpha_{t+1}\mathbb{E}_Z\left[\mathbb{E}_{Y|Z}[Y] \cdot \mathbb{E}_{X|Z}[X]\right] - \theta_*,
$$

$$
B_t = -\left(X_t' X_t^\top - \mathbb{E}_Z\left[\mathbb{E}_{X|Z}[X] \cdot \mathbb{E}_{X|Z}[X]^\top\right]\right)\theta_t + \left(Y_t X_t' - \mathbb{E}_Z\left[\mathbb{E}_{Y|Z}[Y] \cdot \mathbb{E}_{X|Z}[X]\right]\right),
$$

where $A_t$ corresponds to deterministic component, and $B_t$ corresponds to the stochastic component arising due to the use of stochastic gradients. Standard assumptions on the variance of the stochastic gradient made in the stochastic optimization literature include the uniformly bounded variance assumption [Lan20] and the expected smoothness condition [KR20]. In the IVaR setup, such standard assumptions do not hold as $\theta_t$ potentially can be unbounded and thus the gradient estimator can be unbounded. Hence, we establish our results under natural statistical assumptions arising in the context of the IVaR problem, which form the main novelty in our analysis. Furthermore, compared to [MMLR20], notice that we use two samples of $X$ from the conditional distribution $\mathbb{P}_{X|Z}$ and achieve an $\widetilde{\mathcal{O}}(1/T)$ last iterate convergence rate to the global optimal solution, which is the true underlying causal relationship under Assumption 2.1. In comparison, [MMLR20] only provide asymptotic convergence result to the optimal solution of an approximation problem.

**Additional discussion.** It is interesting to explore other losses beyond squared loss (for example to handle classification setting [CF21]), potentially using the Multilevel Monte Carlo (MLMC) based stochastic gradient estimators. While [HCH21], develops such algorithms, the main challenge is about how to avoid mini-batches required in their work leveraging the problem structure in instrumental variable analysis. Furthermore, in the case when $g(\theta; X)$ is parametrized by a non-linear models, for instance, a neural network, we provide local convergence guarantees under additional stronger conditions made typically in the stochastic optimization literature.

**Assumption 2.5.** *Let the following assumptions hold:*

- *Function* $F(\theta)$ *is* $\ell$-*smooth.*

- *The iterates $\{\theta_t\}_{t=1}^{T+1}$ generated by Algorithm 1 are in a compact set A.*
- *The random objects $X|Z$ and $Y|Z$ have bounded variance for any $Z$, i.e., there exist $\sigma > 0$ such that*

$$\mathbb{E}\left[\|X - \mathbb{E}\left[X \mid Z\right]\|^2 \mid Z\right] \leq \sigma^2, \ \mathbb{E}\left[\|Y - \mathbb{E}\left[Y \mid Z\right]\|^2 \mid Z\right] \leq \sigma^2.$$

**Proposition 1.** *Suppose Assumptions 2.1, 2.2, and 2.5 hold. Choosing $\alpha_t \equiv \alpha = \mathcal{O}\left(\frac{1}{\sqrt{T}}\right)$, for Algorithm 1 we have*

$$\min_{1 \leq t \leq T} \mathbb{E}\left[\|\nabla F(\theta_t)\|^2\right] = \mathcal{O}\left(\frac{1}{\sqrt{T}}\right).$$

The proof of the proposition is immediate. Note that under Assumption 2.5, we can deduce that the unbiased gradient estimator $v(\theta) = (g(\theta; X) - Y)\nabla_\theta g(\theta; X')$ has a bounded variance since

$$\begin{aligned}
\text{Var}(v(\theta)) =& \text{Var}(g(\theta; X) - Y)\text{Var}(\nabla_\theta g(\theta; X')) \\
& + \text{Var}(g(\theta; X) - Y)\left(\mathbb{E}\left[\nabla_\theta g(\theta; X')\right]\right)^2 + \text{Var}(\nabla_\theta g(\theta; X'))\left(\mathbb{E}\left[g(\theta; X) - Y\right]\right)^2 \leq \sigma_v^2,
\end{aligned}$$

where the variance and expectation are taken conditioning on $Z$ and $\theta$, and $\sigma_v > 0$ is a constant that only depends on $\sigma$, function $g$ and the compact set $A$ in Assumption 2.5. Then one can directly follow the analysis of non-convex stochastic optimization (see, for example, [GL13, Theorem 2.1]) to obtain Proposition 1. Relaxing the Assumption 2.5 (typically made in the stochastic optimization literature) with more natural assumptions on the statistical model and obtaining a result as in Theorem 1 for the non-convex setting is left as future work.

## 3 One-sample Two-stage Stochastic Gradient Method for IVaR

We now examine designing streaming IVaR algorithm with access to the classical one-sample oracle, i.e., we observe a streaming set of samples $(X_t, Y_t, Z_t)$ at each time point $t$. Note that in this case, using the same $X_t$ (instead of $X_t'$) in (4) makes the stochastic gradient estimator biased.

**Intuition.** Consider the case of linear models, i.e., $Y = \theta_*^\top X + \epsilon_1$ with $X = \gamma_*^\top Z + \epsilon_2$, where $\theta_* \in \mathbb{R}^{d_x \times 1}$, and $\gamma_* \in \mathbb{R}^{d_z \times d_x}$, as also considered in Lemma 1. Recall the true gradient in (3) and the stochastic gradient estimator of Algorithm 1 in (4). Since we no longer have $X_t'$, we replace the term $X_t'$ with the predicted mean of $X_t$ given $Z_t$. Suppose that $\gamma_*$ is known. We specifically replace $\nabla_{\theta_t} g(\theta_t; X_t') = X_t'$ by $\mathbb{E}_{|Z_t}[X_t] = \gamma_*^\top Z_t$. In such a case, indeed we have an unbiased gradient estimator:

$$\begin{aligned}
\mathbb{E}_t\left[\gamma_*^\top Z_t(X_t^\top \theta_t - Y_t)\right] &= \mathbb{E}_t\left[\mathbb{E}_{|Z_t}[X_t]\left(\mathbb{E}_{|Z_t}[X_t]^\top \theta_t - \mathbb{E}_{|Z_t}[Y_t]\right)\right] \\
=& \mathbb{E}_t\left[\gamma_*^\top Z_t Z_t^\top \gamma_*(\theta_t - \theta_*)\right] = \gamma_*^\top \Sigma_Z \gamma_*(\theta_t - \theta_*) = \nabla_\theta F(\theta_t),
\end{aligned}$$

where $\mathbb{E}_t[\cdot]$ is the conditional expectation w.r.t the filtration defined on $\{\gamma_1, \theta_1, \gamma_2, \theta_2, \cdots, \gamma_t, \theta_t\}$.

In reality, $\gamma_*$ is unknown beforehand. Hence, we estimate $\gamma_*$ using some online procedure and replace $\nabla_{\theta_t} g(\theta_t; X_t')$ by $\gamma_t^\top Z_t$ instead of $\gamma_*^\top Z_t$. It leads to the following updates:

$$\theta_{t+1} = \theta_t - \alpha_{t+1}\gamma_t^\top Z_t(X_t^\top \theta_t - Y_t), \qquad \gamma_{t+1} = \gamma_t - \beta_{t+1}Z_t(Z_t^\top \gamma_t - X_t^\top). \tag{11}$$

A closer inspection reveals that the updates in (11) can diverge until $\gamma_t$ is close enough to $\gamma_*$. It is easy to see this fact from the following expansion of $\theta_{t+1} - \theta_*$. We have

$$\begin{aligned}
\theta_{t+1} - \theta_* =& \widehat{Q}_t(\theta_t - \theta_*) + \alpha_{t+1}(\gamma_t - \gamma_*)^\top \Sigma_{ZY} + \alpha_{t+1}D_t\theta_* + \alpha_{t+1}\gamma_t^\top \xi_{Z_t}\gamma_*(\theta_t - \theta_*) \\
& + \alpha_{t+1}\gamma_t^\top \xi_{Z_t}\gamma_*\theta_* + \alpha_{t+1}\gamma_t^\top \xi_{Z_t Y_t} - \alpha_{t+1}\gamma_t^\top Z_t\epsilon_{2,t}^\top \theta_t,
\end{aligned} \tag{12}$$

where

$$\xi_{Z_t} = \Sigma_Z - Z_t Z_t^\top, \quad \xi_{Z_t Y_t} = \Sigma_{ZY} - Z_t Y_t, \quad \widehat{Q}_t := \left(I - \alpha_{t+1}\gamma_t^\top \Sigma_Z \gamma_*\right).$$

However, the matrix $\gamma_t^\top \Sigma_Z \gamma_*$ may not be positive semi-definite, even if $\Sigma_Z$ is positive definite. Thus the negative eigenvalues associated with $\gamma_t^\top \Sigma_Z \gamma_*$ might cause the $\theta_t$ iterates to first diverge, before

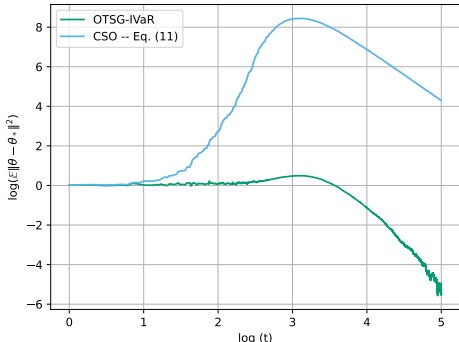

Figure 1: (11) can initially diverge before converging eventually, leading to a worse performance in practical settings compared to Algorithm 2. See Appendix C.2 for the experimental setup.

.

---

**Algorithm 2** One-Sample Two-stage Stochastic Gradient-IVarR (`OTSG-IVaR`)

---

**Input:** Stepsizes $\{\alpha_t\}_t$, $\{\beta_t\}_t$, initial iterates $\gamma_1, \theta_1$.
1: **for** $t = 1, 2, \cdots$ **do**
2:   Sample $Z_t$, sample $X_t$ from $\mathbb{P}_{X|Z_t}$, Sample $Y_t$ from $\mathbb{P}_{Y|X_t}$.
3:   Update

$$\theta_{t+1} = \theta_t - \alpha_{t+1}\gamma_t^\top Z_t(Z_t^\top \gamma_t \theta_t - Y_t), \tag{13}$$

$$\gamma_{t+1} = \gamma_t - \beta_{t+1}Z_t(Z_t^\top \gamma_t - X_t^\top). \tag{14}$$

4: **end for**

---

eventually converging as $\gamma_t$ gets closer to $\gamma_*$. We illustrate this intuition in a simple experiment in Figure 1. To resolve this issue, we propose Algorithm 2, where we replace $g(\theta_t, X_t) = X_t^\top \theta_t$ with $Z_t^T \gamma_t \theta_t$ in (11). With such a modification, in the corresponding decomposition for $\theta_{t+1} - \theta_*$ (see (40)), we have $\widehat{Q}_t = \left(I - \alpha_{t+1}\gamma_t^\top \Sigma_Z \gamma_t\right)$, where the matrix product $\gamma_t^\top \Sigma_Z \gamma_t$ is always positive semi-definite. Hence, with a properly chosen stepsize $\alpha_t$ we could quantify the convergence of $\theta_t$ to $\theta_*$ non-asymptotically. Nevertheless, assuming a warm-start condition on $\theta_0$, we also show the convergence of (11), in Appendix E.3 for completeness.

**Algorithm and Analysis.** Based on the intuition, we present Algorithm 2. One could interpret the algorithm as the SGD analogy of the offline 2SLS algorithm [AI95]. It is also related to the framework of non-linear two-stage stochastic approximation algorithms [DR20, DTSM18, MP06]; albeit the updates of $\theta_t$ and $\gamma_t$ are coupled since both updates use $Z_t$. Furthermore, the dependency between the randomness between the two stages in the IVaR problem, makes the analysis significantly different and more challenging from the classical analysis of two-stage algorithms (see below Theorem 2 for additional details). Finally, while Algorithm 2 is designed for linear models, the intuition behind the method is also applicable to non-linear models (i.e., between $Z$ and $X$, and $X$ and $Y$). We focus on linear models in this work in order to derive our theoretical results. A detailed treatment of the nonlinear case (for which the analysis is significantly nontrivial) is left for future work. We make the following additional assumptions for the convergence analysis of Algorithm 2.

**Assumption 3.1.** *For some constants $C_z, C_{zy} > 0$, we have the following bounds on the fourth moments:*

$$\mathbb{E}\left[\|\Sigma_Z - ZZ^\top\|^4\right] \leq C_z d_z^{\vartheta_5}, \quad \mathbb{E}\left[\|\Sigma_{ZY} - ZY\|^4\right] \leq C_{zy}d_z^{\vartheta_6}, \quad \vartheta := \max\{\vartheta_5, \vartheta_6\}. \tag{15}$$

**Assumption 3.2.** *There exist constants $0 < \mu_Z \leq \lambda_Z < \infty$ such that $\mu_Z I_{d_z} \preceq \Sigma_Z \preceq \lambda_Z I_{d_z}$.*

The above conditions are rather mild moment conditions, similar to Assumption 2.4, and could be easily verified for the linear model setting we consider.

**Assumption 3.3.** *$\{\gamma_t\}_t$ is within a compact set of diameter $C_\gamma d_z^\varkappa$ for some constants $C_\gamma > 0$, $\varkappa \geq 0$.*

We emphasize that Assumption 3.3 is only for the uncoupled sequence $\gamma_t$, which is an SGD sequence for solving a strongly-convex problem. It holds easily in various cases, for example by projecting the iterates onto any compact sets or a sufficiently large ball containing $\gamma^*$. It is also well-known that, without any projection operations, $\{\gamma_t\}_t$ sequence is almost surely bounded [PJ92] under our assumptions. Finally, similar assumptions routinely appear in the analysis of SGD algorithms in various related settings; see, for example, [Tse98, GOP19, HS19, NJN19, AYS20, RGP20].

We now present our result on the convergence of $\{\theta_t\}_t$ below in Theorem 2 (see Appendix E.1 for the proof). In comparison to Theorem 1 (regarding Algorithm 1), we highlight that Theorem 2 provides an any-time guarantee, as the total number of iterations is not required in advance by Algorithm 2.

**Theorem 2.** *Suppose Assumptions 2.3, 2.2 (without $X'_t$), 3.1, 3.3, and 3.2 hold. In Algorithm 2, for any $\iota > 0$, set $\alpha_t = C_\alpha t^{-1+\iota/2}$ and $\beta_t = C_\beta t^{-1+\iota/2}$, where $C_\alpha = \min\{0.5 d_z^{-4\varkappa - \vartheta/2} \lambda_Z^{-1} C_\gamma^{-2}, 0.5(\|\gamma_*\|\lambda_Z)^{-2}\}$, and $C_\beta = \mu^2 d_z^{-1-2\varkappa}/128$. Then, we have*

$$\mathbb{E}\left[\|\theta_t - \theta^*\|^2\right] = O\left(\frac{1}{t^{1-\iota}}\right).$$

**Remark 1.** *In Theorem 2, we present the step-size choices for the fastest rate of convergence. In the proof of Theorem 2 (see Appendix E.1), we show that convergence can be guaranteed for a range of step-sizes given by $\alpha_t = C_\alpha t^{-a}$, $\beta_t = C_\beta t^{-b}$, where $1/2 < a, b < 1$, $b > 2 - 2a$ with corresponding rate being $\mathbb{E}\left[\|\theta_t - \theta^*\|^2\right] = O(\max\{t^{-b(2-(1-\iota/2)^{-1})}, t^{-a}\log(2/\iota - 1)\})$. In particular, one requires $a, b < 1$ to ensure $(\alpha_t - \alpha_{t+1})/\alpha_t = o(\alpha_t)$, and $(\beta_t - \beta_{t+1})/\beta_t = o(\beta_t)$, as is standard in stochastic approximation literature (see, for example, [CLTZ20, PJ92]).*

**Proof Techniques.** The major challenge towards the convergence analysis of $\{\theta_t\}_t$ lies in the interaction term $\gamma_t Z_t Z_t^\top \gamma_t \theta_t$ between $\gamma_t$ and $\theta_t$ in (13). This multiplicative interaction term leads to an involved dependence between the noise in the stochastic gradient updates for the two stages. Such a dependence has not been considered in existing analysis of non-linear two time-scale algorithms [MP06, MSB$^+$09, DTSM18, DR20, XL21, WZZ21, Doa22]. In addition, [Doa22] considers the case when the noise sequence is not only independent of each other but also independent of iterate locations. Furthermore, they assumes (see their Assumption 3) that the condition in Assumption 2.3 holds for all $\gamma$ whereas Assumption 2.3 only needs to hold for $\gamma_*$, that is much milder. Similarly, many works (for example, Assumption 1 in [WZZ21], Assumption 2 in [XL21] and Theorem 2 in [MSB$^+$09]) assume that the iterates of both stages are bounded in a compact set and consequently, and hence the variance of the stochastic gradients are also uniformly bounded.

In our setting, firstly, the stochastic gradient in (13), evaluated at $(\theta_t, \gamma_t)$ is biased:

$$\mathbb{E}_{t,Z_t}\left[\gamma_t^\top Z_t(Z_t^\top \gamma_t \theta_t - Y_t)\right] = \mathbb{E}_{t,Z_t}\left[\gamma_t^\top Z_t(Z_t^\top \gamma_t \theta_t - Z_t^\top \gamma_* \theta_*)\right] = \mathbb{E}_t\left[\gamma_t^\top \Sigma_Z(\gamma_t \theta_t - \gamma_* \theta_*)\right]$$
$$= \gamma_t^\top \Sigma_Z \gamma_t(\theta_t - \theta_*) + \gamma_t^\top \Sigma_Z(\gamma_t - \gamma_*)\theta_* \neq \gamma_*^\top \Sigma_Z \gamma_*(\theta_t - \theta_*) = \nabla_\theta F(\theta_t).$$

Furthermore, even under Assumption 3.3, the variance of the stochastic gradient is not (13) uniformly bounded. Overcoming these issues, in addition to the aforementioned dependence between the noise in the stochastic gradient updates for the two stages, forms the major novelty in our analysis. We proceed by noting that if $\gamma_*$, $\Sigma_Z$, and $\Sigma_{ZY}$ were known beforehand, one conduct deterministic gradient updates, i.e., $\widetilde{\theta}_{t+1} = \widetilde{\theta}_t - \alpha_{t+1} \gamma_*^\top \left(\Sigma_Z \gamma_* \widetilde{\theta}_t - \Sigma_{ZY}\right)$, to obtain $\theta_*$. By standard results on gradient descent for strongly convex functions (see, for example, [Nes13]), $\{\widetilde{\theta}_t\}_t$ converges exponentially fast as stated in Lemma 4. Hence, it remains to show that the trajectory of $\theta_t$ converges to the trajectory of $\widetilde{\theta}_t$. That is, defining the sequence $\delta_t := \theta_t - \widetilde{\theta}_t$, our goal is to establish the convergence rate of $\mathbb{E}\left[\|\delta_t\|_2^2\right]$. We first provide an intermediate bound (see Lemma 6) and then progressively sharpen to a tighter bound (see Lemma 7). In doing so, it is also required to show that $\mathbb{E}\left[\|\theta_t\|^4\right]$ is bounded, which we prove in Lemma 5. The proof of Lemma 5 is non-trivial and requires carefully chosen stepsizes satisfying $\sum_{t=1}^\infty (\alpha_t^2 + \alpha_t \sqrt{\beta_t}) < \infty$.

## 4 Numerical Experiments

**Experiments for Algorithm 1 (TOSG-IVaR).** We first consider the following problem, in which $(Z, X, Y)$ is generated via

$$Z \sim \mathcal{N}(0, I_{d_z}), \ X = \phi(\gamma_*^\top Z) + c \cdot (h + \epsilon_x), \ Y = \theta_*^\top X + c \cdot (h_1 + \epsilon_y),$$

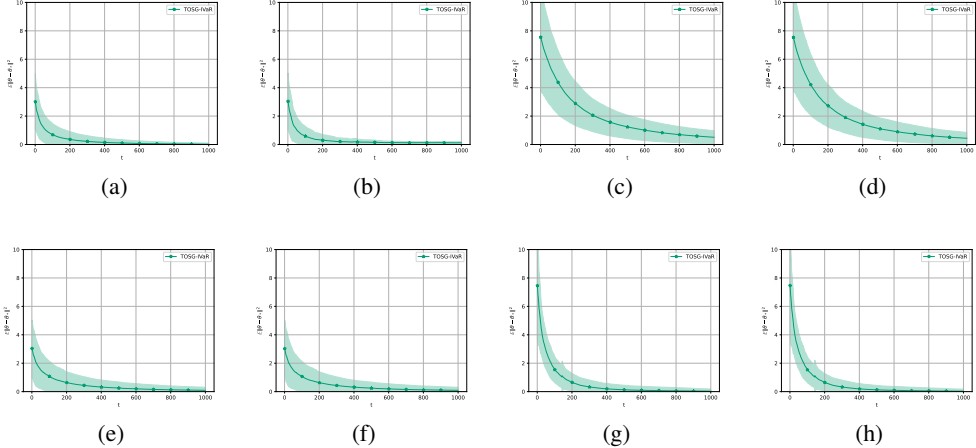

Figure 2: $\mathbb{E}[\|\theta_t - \theta_*\|^2]$ of Algorithm 1 under different settings detailed in Section 4.

where $c > 0$ is a scalar to control the variance of the noise vector, and $h_1$ is the first coordinate of $h$. The noise vectors (or scalar) $h, \epsilon_x, \epsilon_y$ are independent of $Z$, and we have $h \sim \mathcal{N}(\mathbf{1}_{d_x}, I_{d_x})$, $\epsilon_x \sim \mathcal{N}(0, I_{d_x}), \epsilon_y \sim \mathcal{N}(0, 1)$. In each iteration, one tuple $(X, X', Y)$ is generated and used to update $\theta_t$ according to Algorithm 1. We set $(d_x, d_z) \in \{(4, 8), (8, 16)\}$, $c \in \{0.1, 1.0\}$, and $\phi(s) \in \{s, s^2\}$. We repeat each setting 50 times and report the curves of $\mathbb{E}[\|\theta_t - \theta_*\|^2]$ in Figure 2, where the expectation is computed as the average of $\|\theta_t - \theta_*\|^2$ of all trials, and the shaded region represents the standard deviation. The first row and the second row correspond to $\phi(s) = s$ and $\phi(s) = s^2$ respectively. Here, $c = 0.1$ for odd columns and $c = 1.0$ for even columns. We have $(d_x, d_z) = (4, 8)$ for the first two columns and $(d_x, d_z) = (8, 16)$ for the last two columns. Empirically, we can observe that our Algorithm 1 performs well across all different settings.

**Experiments for Algorithm 2 (`OTSG-IVaR`).** Next, we compare our Algorithm 2 as well as its variant and Algorithm 1 in [DVB24]. We write "OTSG-IVaR", "CSO – Eq. (11)" and "[DVB23]" to represent Algorithm 2, Algorithm 2 with the updates replaced by (11) and Algorithm 1 in [DVB24] (see Appendix A). We follow simulation settings similar to [DVB24]:

$$Y = \theta_*^\top X + \nu, \qquad X = \gamma_*^\top Z + \epsilon, \qquad \epsilon = \sigma_\epsilon \mathcal{N}(0, I_{d_x}), \qquad \nu = \rho\epsilon_1 + \mathcal{N}(0, 0.25), \quad (16)$$

where $\epsilon_1$ is the first coordinate of $\epsilon$, $\theta_* \in \mathbb{R}^{d_x}$ is a unit vector chosen uniformly randomly, and $\gamma_* \in \mathbb{R}^{d_z \times d_x}$ where $\gamma_{ij} = 0$ for $i \neq j$, and $\gamma_{ij} = 1$ for $i = j$, $i = 1, 2, \cdots, d_x$, and $j = 1, 2, \cdots, d_z$. Here $\rho$ controls the level of endogeneity in the model. We compare the performance of Algorithm 2 with (11), and O2SLS [DVB24] for $\rho = 1, 4$, and $\sigma_\epsilon = 0.5, 1$. By varying $\sigma_\epsilon$ we control the correlation between $X$ and $Z$. We consider two settings $(d_x, d_z) = (1, 1)$, and $(d_x, d_z) = (8, 16)$. As performance metric, in Figure 3 we plot $\mathbb{E}\left[\|\theta_t - \theta_*\|^2\right]$ where the $\mathbb{E}\left[\cdot\right]$ is approximated by averaging over 50 trials, and both axes are in $\log$ scale (base 10). We also show, in Figure 4, the convergence of the test Mean Squared Error (MSE) evaluated over 400 test samples to the best possible test MSE where $\theta_*$ and $\gamma_*$ are known beforehand. For Figures 3 and 4, the first row and second row corresponds to $(d_x, d_z) = (1, 1)$ and $(d_x, d_z) = (8, 16)$ respectively, and $\sigma_\epsilon = 0.5$ in odd columns and $\sigma_\epsilon = 1.0$ in even columns. We have $\rho = 1.0$ for the first two columns and $\rho = 4.0$ for the last two columns. We can observe that O2SLS has much larger variance in different settings, while our algorithms perform consistently well in all settings. We further conduct experiments on real-world datasets provided in [AE96] and [Rya12]. Due to space limit, we include the numerical results in Section C.3 of the Appendix.

## 5   Conclusion

We presented streaming algorithms for least-squares IVaR based on directly solving the associated conditional stochastic optimization formulation in (2). Our algorithms have several benefits, including avoidance of mini-batches and matrix inverses. We show that the expected rates of convergences for the proposed algorithms are of order $\mathcal{O}(\log T/T)$ and $\mathcal{O}(1/T^{1-\iota})$, for any $\iota > 0$, under the

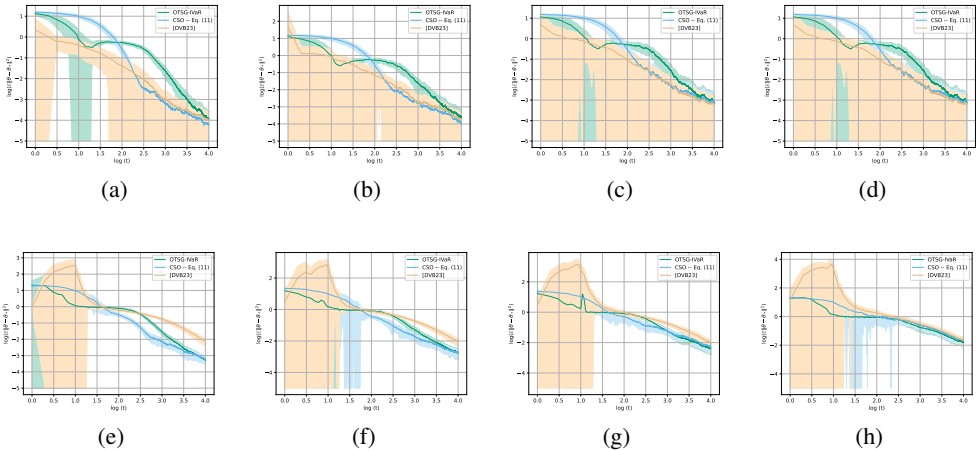

Figure 3: Comparison of $\mathbb{E}[\|\theta_t - \theta_*\|^2]$ (log-log scale) for Algorithm 2, Eq. 11 and [DVB24].

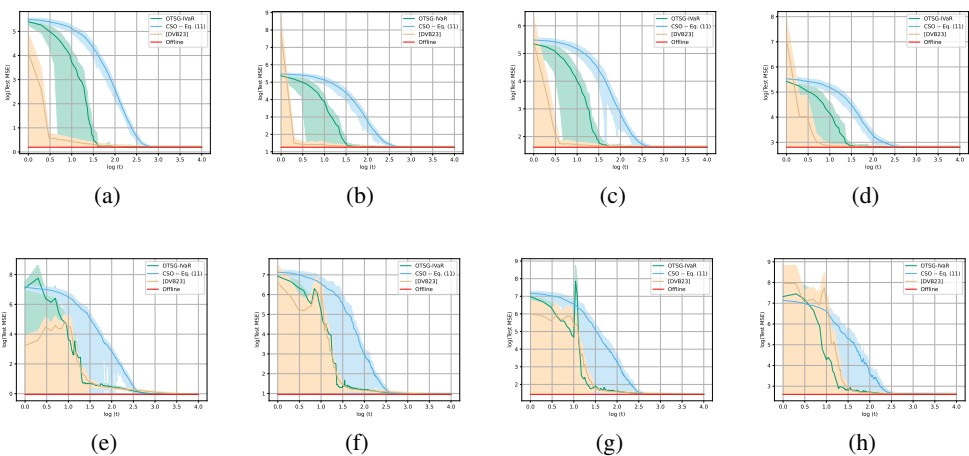

Figure 4: Comparison of test MSE (log-log scale) for Algorithm 2, Eq. 11 and [DVB24].

availability of two-sample and one-sample oracles, respectively. As future work, it is interesting to develop streaming inferential methods for IVaR. Leveraging related works for the vanilla SGD [PJ92, ABE19, SZ22, CLTZ20, ZCW23] to the setting of Algorithms 1 and 2, provides a concrete direction to establish Central Limit Theorems and develop limiting covariance estimation procedures.

## Acknowledgments and Disclosure of Funding

YH was supported as a part of NCCR Automation, a National Centre of Competence (or Excellence) in Research, funded by the Swiss National Science Foundation (grant number 51NF40_225155). KB was supported in part by NSF grants DMS-2053918 and DMS-2413426.

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

# Appendix

## A  Online updates of [DVB24]

For the sake of clarity, we present the O2SLS algorithm proposed in [DVB24, v3][1] in the streaming format, without any explicit matrix inversions that we used in our experiments:

$$\theta_{t+1} = (I - U_t\gamma_t{}^\top Z_t Z_t^\top \gamma_t)\theta_t + U_t\gamma_t{}^\top Z_t Y_t$$
$$\gamma_{t+1} = (I - V_t Z_t Z_t^\top)\gamma_t + V_t Z_t X_t^\top$$
$$U_{t+1} = U_t - \frac{U_t\gamma_t{}^\top Z_t Z_t^\top \gamma_t U_t}{1 + Z_t^\top \gamma_t U_t \gamma_t{}^\top Z_t}$$
$$V_{t+1} = V_t - \frac{V_t Z_t Z_t^\top V_t}{1 + Z_t^\top V_t Z_t} \qquad V_0 = \lambda^{-1} I_{d_z},$$

where $U_t, V_t$ are two additional matrix sequences which tracks the matrix inverse of $\sum_{i=1}^{t} \gamma_i^\top Z_i Z_i^\top \gamma_i$, and $(\lambda I_{d_z} + \sum_{i=1}^{t} Z_t Z_t^\top)$ respectively for a user defined parameter $\lambda$. As mentioned in [DVB24], we choose $\lambda = 0.1$. The major difference between O2SLS and Algorithm 2 is that O2SLS takes an online two-stage regression approach to minimize a suitably defined regret whereas we take a conditional stochastic optimization point of view which requires carefully chosen step-sizes. In our Algorithm 2, we do not need to explicitly or implicitly do matrix inverse which can potentially cause stability issues. Furthermore, unlike [DVB24], we neither assume $\sum_{i=1}^{t} Z_i Z_i^\top$ is invertible for all $t$ nor do we assume that $Z$ is a bounded random variable for our analysis. Finally, the per-iteration computational complexity and memory requirement of Algorithm 2 is significantly better than O2SLS; see Section B.

## B  Per-iteration Complexities

For the linear case, i.e., the underlying relationship between $Z$ and $X$ as well as $X$ and $Y$ are linear, Table 1 summarizes the per-iteration memory costs and number of arithmetic operations of the original O2SLS [DVB23], the updated O2SLS [DVB24] that we provide a matrix form update in Appendix A, TOSG-IVaR (Alg 1), and OTSG-IVaR (Alg 2) at the $t$-th iteration.

Notice that the original version of O2SLS [DVB23] has a per-iteration and memory cost dependent on the iteration number $t$ as it needs to use all the samples accumulated till the iteration $t$ to conduct an offline 2SLS at each iteration. The updated O2SLS [DVB24] (the algorithm that we compare to) uses samples obtained at iteration $t$ to perform the update. Although the updated O2SLS avoids explicit matrix inversion, it is obvious that its arithmetic operations and memory cost per iteration are larger than our TOSG-IVaR and OTSG-IVaR.

We highlight that the TOSG-IVaR, which uses two samples $X$ and $X'$ from the conditional distribution $\mathbb{P}(X \mid Z)$, requires only $\mathcal{O}(d_x)$ memory and arithmetic operations at each iteration.

For a fair comparison, we assume that two $n \times n$ matrices multiplication admits an $\mathcal{O}(n^3)$ complexity, i.e., using normal textbook matrix multiplication. We also assume computing the inversion of a $n \times n$ matrix admits an $\mathcal{O}(n^3)$ complexity. Interested readers may refer to [Pap03] for more details about faster algorithms with better complexities for matrix operations.

---

[1]Note that the streaming algorithm was not present in version 1, i.e., [DVB23, v1].

Table 1: Memory cost and the number of arithmetic operations at iteration $t$.

| Algorithm | Memory cost | Arithmetic Operations |
|---|---|---|
| O2SLS [DVB23, v1] | $t(d_x + d_z) + d_z d_x + d_x$ | $\mathcal{O}(d_x^3 + t d_x^2 + t d_x d_z)$ |
| O2SLS [DVB24, v3] (Sec. A) | $d_x^2 + d_z^2 + d_z d_x + d_x$ | $\mathcal{O}(d_x^2 + d_z^2 + d_z d_x)$ |
| TOSG-IVaR (our Alg 1) | $d_x$ | $\mathcal{O}(d_x)$ |
| OTSG-IVaR (our Alg 2) | $d_x d_z + d_x$ | $\mathcal{O}(\min(d_x^2, d_z^2) + d_z d_x)$ |

## C  Experimental Details

### C.1  Compute Resources

All experiments in Section 4 were conducted on a computer with an 11th Intel(R) Core(TM) i7-11370H CPU. The time and space required to run our experiments are negligible and we anticipate they can be conducted in almost all computers.

### C.2  Experimental Details for Figure 1

In Figure 1, we show an example where the updates (11) may diverge first before converging eventually and finite time performance can be much worse compared to Algorithm 2. For this experiment, we choose the model presented in (16) with $d_x = d_z = 1$, $\theta_* = 1$, $\gamma_* = -1$, $\rho = 4$, and $\sigma_\epsilon = 1$. When initialized at $\gamma_0 = 10$, and $\theta_0 = 0$, the updates in (11) keeps diverging rapidly at first whereas Algorithm 2 is much more stable. So, by the end of $100,000$ iterations, while Algorithm 2 achieves an error of $\approx 10^{-5}$, (11) achieves $\approx 10^4$ that is worse than it was at initialization because (11) has not recovered from the initial divergence phase yet. However, once (11) starts converging, the convergence rate of (11) is similar to Algorithm 2 as one can see from Figure 1 (also see our discussion on the convergence of (11) in Section E.3).

### C.3  Additional Experiments on Real-World Dataset

In this section we provide experimental results on real-world datasets provided in [AE96] and [Rya12]. The results are included in Figures 5 and 6 respectively. Following the convention in Section 4, we write "OTSG-IVaR", "CSO – Eq. (11)" and "[DVB23]" to represent Algorithm 2, Algorithm 2 with the updates replaced by (11) and Algorithm 1 in [DVB24].

#### C.3.1  Children and Their Parents' Labor Supply Data in [AE96]

The outcome $Y$ is number of working weeks divided by 52, the regressor $X$ is $\mathbf{1}$(number of children is greater than 2), and the instrumental variable $Z$ is $\mathbf{1}$(first two siblings are of same sex), where $\mathbf{1}(\cdot)$ is the indicator function. At each time we randomly sample a data-point from this dataset without replacement. Since the "true" parameter $\theta_*$ is unknown in real data, we use the offline model parameter estimate as our ground truth, following [CLL+23]. We include the results in Figure 5, in which we observe that CSO performs similar to [DVB23] whereas OTSG-IVaR performs much better in terms of convergence speed. Moreover, the estimation errors of CSO and [DVB23] plateau after $\approx 10,000$ iterations whereas OTSG-IVaR keeps on improving the estimate over the observed horizon.

#### C.3.2  U.S. Portland Cement Industry Data in [Rya12]

Response variable $Y$ is $\log$(shipped), and the predictor $X$ is $\log$(price). There are 4 instrumental variables given by wage in dollars per hour for skilled manufacturing workers, electricity price, coal price, and gas price. Unlike the previous dataset, we have only 483 data samples here. So, to mimic an i.i.d. data stream, we divide our training into multiple epochs of length equal to the number of training data samples, and over each epoch, at each iteration, we sample one data point uniformly randomly without replacement from the training data to generate the data-stream. Both OTSG-IVaR and CSO perform much better than [DVB23] in terms of convergence speed. Figure 6(b) is a magnified view of iterations $> 50$ to highlight the performance difference between various algorithms.

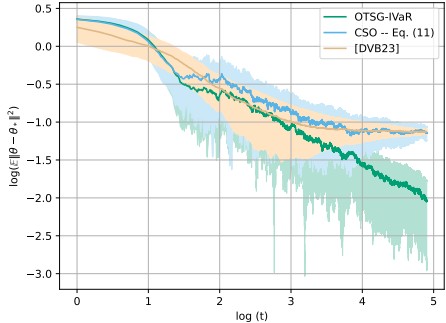

Figure 5: Comparison of $\mathbb{E}[\|\theta_t - \theta_*\|^2]$ (log-log scale) for Algorithm 2, Eq. 11 and [DVB24].

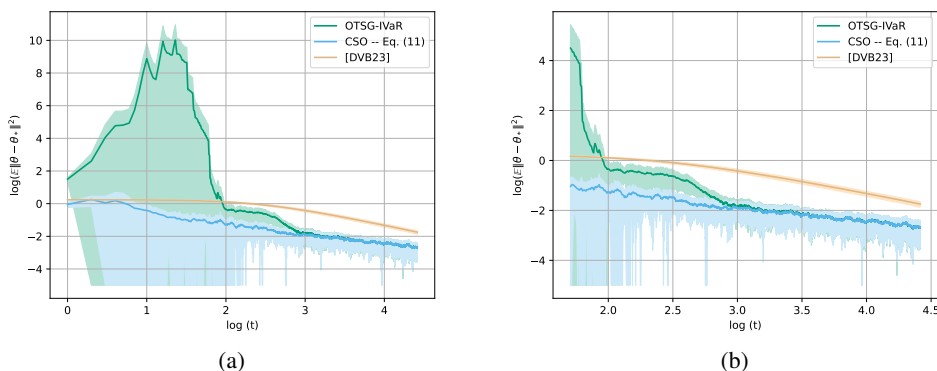

(a)                                              (b)

Figure 6: Comparison of $\mathbb{E}[\|\theta_t - \theta_*\|^2]$ (log-log scale) for Algorithm 2, Eq. 11 and [DVB24].

# D   Proofs for Section 2

## D.1   Proof of Theorem 1

*Proof.* We aim to find the optimal $\theta_*$. According to (2), we know

$$\mathbb{E}_Z\Big[\mathbb{E}_{X|Z}[X] \cdot \mathbb{E}_{X|Z}[X]^\top\Big]\theta_* = \mathbb{E}_Z\Big[\mathbb{E}_{Y|Z}[Y] \cdot \mathbb{E}_{X|Z}[X]\Big] \tag{17}$$

The updates in Algorithm 1 can be written as

$$\theta_{t+1} = \theta_t - \alpha_{t+1}(X_t^\top \theta_t - Y_t)X_t^\top.$$

Hence we have

$$
\begin{aligned}
&\theta_{t+1} - \theta_* \\
=&\theta_t - \alpha_{t+1}\mathbb{E}_Z\Big[\mathbb{E}_{X|Z}[X] \cdot \mathbb{E}_{X|Z}[X]^\top\Big]\theta_t + \alpha_{t+1}\mathbb{E}_Z\Big[\mathbb{E}_{Y|Z}[Y] \cdot \mathbb{E}_{X|Z}[X]\Big] - \theta_* \\
&- \alpha_{t+1}\Big(X_t'X_t^\top - \mathbb{E}_Z\Big[\mathbb{E}_{X|Z}[X] \cdot \mathbb{E}_{X|Z}[X]^\top\Big]\Big)\theta_t + \alpha_{t+1}\Big(Y_tX_t' - \mathbb{E}_Z\Big[\mathbb{E}_{Y|Z}[Y] \cdot \mathbb{E}_{X|Z}[X]\Big]\Big).
\end{aligned}
\tag{18}
$$

Now we analyze the convergence and variance separately. For the convergence part, we have

$$
\begin{aligned}
&\theta_t - \alpha_{t+1}\mathbb{E}_Z\Big[\mathbb{E}_{X|Z}[X] \cdot \mathbb{E}_{X|Z}[X]^\top\Big]\theta_t + \alpha_{t+1}\mathbb{E}_Z\Big[\mathbb{E}_{Y|Z}[Y] \cdot \mathbb{E}_{X|Z}[X]\Big] - \theta_* \\
=&\Big(I - \alpha_{t+1}\mathbb{E}_Z\Big[\mathbb{E}_{X|Z}[X] \cdot \mathbb{E}_{X|Z}[X]^\top\Big]\Big)(\theta_t - \theta_*).
\end{aligned}
\tag{19}
$$

For the variance part we have

$$\mathbb{E}\Big[\|X_t'X_t^\top - \mathbb{E}_Z\big[\mathbb{E}_{X|Z}[X]\cdot\mathbb{E}_{X|Z}[X]^\top\big]\|^2\Big]$$

$$=\mathbb{E}\Big[\|X_t'X_t^\top - \mathbb{E}_{X|Z_t}[X]\cdot\mathbb{E}_{X|Z_t}[X]^\top + \mathbb{E}_{X|Z_t}[X]\cdot\mathbb{E}_{X|Z_t}[X]^\top - \mathbb{E}_Z\big[\mathbb{E}_{X|Z}[X]\cdot\mathbb{E}_{X|Z}[X]^\top\big]\|^2\Big]$$

$$\leq 2\mathbb{E}\Big[\|X_t'X_t^\top - \mathbb{E}_{X|Z_t}[X]\cdot\mathbb{E}_{X|Z_t}[X]^\top\|^2 + \|\mathbb{E}_{X|Z_t}[X]\cdot\mathbb{E}_{X|Z_t}[X]^\top - \mathbb{E}_Z\big[\mathbb{E}_{X|Z}[X]\cdot\mathbb{E}_{X|Z}[X]^\top\big]\|^2\Big]$$

$$\leq 2C_x d_x^{\vartheta_1} + 2C_{xx} d_z^{\vartheta_3} =: \sigma_1^2 \tag{20}$$

Similarly, we have

$$\mathbb{E}\Big[\|Y_t X_t' - \mathbb{E}_Z\big[\mathbb{E}_{Y|Z}[Y]\cdot\mathbb{E}_{X|Z}[X]\big]\|^2\Big]$$

$$=\mathbb{E}\Big[\|Y_t X_t' - \mathbb{E}_{Y|Z_t}[Y]\cdot\mathbb{E}_{X|Z_t}[X]\|^2 + \|\mathbb{E}_{Y|Z_t}[Y]\cdot\mathbb{E}_{X|Z_t}[X] - \mathbb{E}_Z\big[\mathbb{E}_{Y|Z}[Y]\cdot\mathbb{E}_{X|Z}[X]\big]\|^2\Big]$$

$$\leq C_y d_x^{\vartheta_2} + C_{yx} d_z^{\vartheta_4} =: \sigma_2^2. \tag{21}$$

Now we know from (18), (19), (20), and (21) that

$$\|\theta_{t+1} - \theta_*\|^2 = \|A_t\|^2 + 2\alpha_{t+1}\langle A_t, B_t\rangle + \alpha_{t+1}^2\|B_t\|^2. \tag{22}$$

where

$$A_t = \Big(I - \alpha_{t+1}\mathbb{E}_Z\big[\mathbb{E}_{X|Z}[X]\cdot\mathbb{E}_{X|Z}[X]^\top\big]\Big)(\theta_t - \theta_*)$$

$$B_t = -\Big(X_t'X_t^\top - \mathbb{E}_Z\big[\mathbb{E}_{X|Z}[X]\cdot\mathbb{E}_{X|Z}[X]^\top\big]\Big)\theta_t + \Big(Y_t X_t' - \mathbb{E}_Z\big[\mathbb{E}_{Y|Z}[Y]\cdot\mathbb{E}_{X|Z}[X]\big]\Big).$$

This implies

$$\mathbb{E}_{\theta_{t+1}|\theta_t}\Big[\|\theta_{t+1} - \theta_*\|^2\Big]$$

$$=\|\Big(I - \alpha_{t+1}\mathbb{E}_Z\big[\mathbb{E}_{X|Z}[X]\cdot\mathbb{E}_{X|Z}[X]^\top\big]\Big)(\theta_t - \theta_*)\|^2$$

$$+ \alpha_{t+1}^2\mathbb{E}_{X_t,X_t',Y_t,Z_t|\theta_t}\Big[\|\Big(X_t'X_t^\top - \mathbb{E}_Z\big[\mathbb{E}_{X|Z}[X]\cdot\mathbb{E}_{X|Z}[X]^\top\big]\Big)\theta_t - \Big(Y_t X_t' - \mathbb{E}_Z\big[\mathbb{E}_{Y|Z}[Y]\cdot\mathbb{E}_{X|Z}[X]\big]\Big)\|^2\Big]$$

$$\leq (1 - \alpha_{t+1}\mu)^2\|\theta_t - \theta_*\|^2 + 3\alpha_{t+1}^2\Big(\sigma_1^2\|\theta_t - \theta_*\|^2 + \sigma_1^2\|\theta_*\|^2 + \sigma_2^2\|\theta_*\|^2\Big)$$

$$\leq ((1 - \alpha_{t+1}\mu)^2 + 3\alpha_{t+1}^2\sigma_1^2)\|\theta_t - \theta_*\|^2 + 3\alpha_{t+1}^2\sigma_1^2\|\theta_*\|^2 + 3\alpha_{t+1}^2\sigma_2^2\|\theta_*\|^2, \tag{23}$$

where the first inequality uses Cauchy-Schwarz inequality, the definition of $\sigma_1, \sigma_2$ and Assumption 2.4. Choosing $\alpha_{t+1}$ such that

$$((1 - \alpha_{t+1}\mu)^2 + 3\alpha_{t+1}^2\sigma_1^2) \leq 1 - \alpha_{t+1}\mu \Leftrightarrow \alpha \leq \frac{\mu}{\mu^2 + 3\sigma_1^2}$$

and taking expectation on both sides of (23), we have

$$\mathbb{E}\Big[\|\theta_{t+1} - \theta_*\|^2\Big] \leq (1 - \alpha_{t+1}\mu)\mathbb{E}\Big[\|\theta_t - \theta_*\|^2\Big] + 3\alpha_{t+1}^2\sigma_1^2\|\theta_*\|^2 + 3\alpha_{t+1}^2\sigma_2^2\|\theta_*\|^2.$$

Now, we use the following result.

**Lemma 2.** *Suppose we have three sequences $\{a_t\}_{t=0}^\infty, \{b_t\}_{t=0}^\infty, \{r_t\}_{t=0}^\infty$ satisfying*

$$a_{t+1} \leq r_t a_t + b_t, r_t > 0 \tag{24}$$

*for any $t \geq 0$. Define $R_{t+1} = \prod_{i=0}^t r_i$, we have*

$$a_{t+1} \leq R_{t+1} a_0 + \sum_{i=0}^t \frac{R_{t+1} b_i}{R_{i+1}}.$$

By Lemma 2, we know

$$\mathbb{E}\Big[\|\theta_{t+1} - \theta_*\|^2\Big] \leq \prod_{i=0}^t (1 - \alpha_i\mu)\mathbb{E}\Big[\|\theta_0 - \theta_*\|^2\Big] + (3\sigma_1^2\|\theta_*\|^2 + 3\sigma_2^2\|\theta_*\|^2)\sum_{i=0}^t \alpha_i^2 \prod_{j=i+1}^t (1 - \alpha_j\mu).$$

Now if we set $\alpha_i = \alpha$, we know

$$\mathbb{E}\Big[\|\theta_t - \theta_*\|^2\Big] \leq (1 - \alpha\mu)^t \mathbb{E}\Big[\|\theta_0 - \theta_*\|^2\Big] + \alpha^2\Big(\sum_{i=0}^{t}(1 - \alpha\mu)^i\Big)(3\sigma_1^2\|\theta_*\|^2 + 3\sigma_2^2\|\theta_*\|^2)$$

$$\leq e^{-t\alpha\mu}\mathbb{E}\Big[\|\theta_0 - \theta_*\|^2\Big] + \frac{\alpha}{\mu}(3\sigma_1^2\|\theta_*\|^2 + 3\sigma_2^2\|\theta_*\|^2)$$

Choosing $\alpha, T$ such that $\alpha = \frac{\log T}{\mu T} \leq \frac{\mu}{\mu^2 + 3\sigma_1^2}$, we know

$$\mathbb{E}\Big[\|\theta_T - \theta_*\|^2\Big] \leq \frac{\mathbb{E}\Big[\|\theta_0 - \theta_*\|^2\Big]}{T} + \frac{3\|\theta_*\|^2(\sigma_1^2 + \sigma_2^2)\log T}{\mu^2 T}.$$

$\square$

## D.2 Proof of Lemma 2

*Proof.* We notice from (24) that for any $0 \leq i \leq t$, we have

$$\frac{a_{i+1}}{R_{i+1}} \leq \frac{a_i}{R_i} + \frac{b_i}{R_{i+1}}.$$

Taking summation on both sides, we have

$$\frac{a_{t+1}}{R_{t+1}} \leq \frac{a_0}{R_0} + \sum_{i=0}^{t}\frac{b_i}{R_{i+1}}$$

which completes the proof by multiplying $R_{t+1}$ on both sides. $\square$

## D.3 Proof of Lemma 1

*Proof.* We first notice that Assumption 2.3 holds since

$$\mathbb{E}_Z\Big[\mathbb{E}_{X|Z}[X] \cdot \mathbb{E}_{X|Z}[X]^\top\Big] = \mathbb{E}_Z\Big[\phi(\gamma_*^\top Z) \cdot \phi(\gamma_*^\top Z)^\top\Big] \succeq \mu I.$$

For (5) and (6), we have

$$\mathbb{E}\Big[\|X'X^\top - \mathbb{E}_{X|Z}[X]\mathbb{E}_{X|Z}[X]^\top\|^2\Big]$$

$$=\mathbb{E}\Big[\|\epsilon_2'\phi(\gamma_*^\top Z)^\top + \phi(\gamma_*^\top Z)\epsilon_2^\top + \epsilon_2'\epsilon_2^\top\|^2\Big]$$

$$\leq 3\mathbb{E}\Big[\|\epsilon_2'\phi(\gamma_*^\top Z)^\top\|^2 + \|\phi(\gamma_*^\top Z)\epsilon_2^\top\|^2 + \|\epsilon_2'\epsilon_2^\top\|^2\Big]$$

$$=3\mathbb{E}\Big[\|\phi(\gamma_*^\top Z)\epsilon_2'^\top\epsilon_2'\phi(\gamma_*^\top Z)^\top\| + \|\phi(\gamma_*^\top Z)\epsilon_2^\top\epsilon_2\phi(\gamma_*^\top Z)^\top\| + |\epsilon_2^\top\epsilon_2'|^2\Big] = \mathcal{O}(d_x^2), \quad (25)$$

and

$$\mathbb{E}\Big[\|YX' - \mathbb{E}_{Y|Z}[Y]\mathbb{E}_{X|Z}[X]\|^2\Big]$$

$$=\mathbb{E}\Big[\|X'X^\top\theta_* + \epsilon_1X' - \mathbb{E}_{X|Z}[X]\mathbb{E}_{X|Z}[X]^\top\theta_*\|^2\Big]$$

$$\leq 2\mathbb{E}\Big[\|X'X^\top\theta_* - \mathbb{E}_{X|Z}[X]\mathbb{E}_{X|Z}[X]^\top\theta_*\|^2\Big] + 2\mathbb{E}\Big[\epsilon_1^2\|\phi(\gamma_*^\top Z) + \epsilon_2'\|^2\Big]$$

$$=\mathcal{O}(\|\theta_*\|^2\sigma_{\epsilon_2}^2d_x^2 + \sigma_{\epsilon_1}^2d_x + \sigma_{\epsilon_1,\epsilon_2}^2d_x),$$

where the first inequality uses Cauchy-Schwarz inequality, and the second equality uses (9), (10) and (25). For (7) we have

$$\mathbb{E}\Big[\|\mathbb{E}_{X|Z}[X] \cdot \mathbb{E}_{X|Z}[X]^\top - \mathbb{E}_Z\Big[\mathbb{E}_{X|Z}[X] \cdot \mathbb{E}_{X|Z}[X]^\top\Big]\|^2\Big]$$

$$=\mathbb{E}\Big[\|\phi(\gamma_*^\top Z)\phi(\gamma_*^\top Z)^\top - \mathbb{E}\big[\phi(\gamma_*^\top Z)\phi(\gamma_*^\top Z)^\top\big]\|^2\Big] = \mathcal{O}(d_z)$$

where the last equality uses (10). Using the above conclusion in (8), we have

$$\mathbb{E}\Big[\|\mathbb{E}_{Y|Z}[Y] \cdot \mathbb{E}_{X|Z}[X] - \mathbb{E}_Z\Big[\mathbb{E}_{Y|Z}[Y] \cdot \mathbb{E}_{X|Z}[X]\Big]\|^2\Big]$$

$$=\mathbb{E}\Big[\|\mathbb{E}_{X|Z}[X] \cdot \mathbb{E}_{X|Z}[X]^\top\theta_* - \mathbb{E}_Z\Big[\mathbb{E}_{X|Z}[X] \cdot \mathbb{E}_{X|Z}[X]^\top\theta_*\Big]\|^2\Big] = \mathcal{O}(\|\theta_*\|^2 d_z).$$

$\square$

# E  Proofs for Section 3

## E.1  Proof of Theorem 2

*Proof of Theorem 2 .* Recall that $\xi_{Z_t}$, and $\xi_{Z_t Y_t}$ are the i.i.d. noise sequences

$$\xi_{Z_t} = \Sigma_Z - Z_t Z_t^\top,$$
$$\xi_{Z_t Y_t} = \Sigma_{ZY} - Z_t Y_t.$$

Note $\gamma_*$, and $\theta_*$ can be written as $\gamma_* = \Sigma_Z^{-1} \Sigma_{ZX} \in \mathbb{R}^{d_z \times d_x}$, and $\theta_* = \left(\gamma_*^\top \Sigma_z \gamma_*\right)^{-1} \gamma_*^\top \Sigma_{ZY} \in \mathbb{R}^{d_x}$ which we are going to use throughout the proof.

To quantify the bias, we use the following bound on $\mathbb{E}\left[\|\gamma_t - \gamma_*\|_2^k\right]$, $k = 1, 2, 4$, proved in Lemma 3.2 of [CLTZ20].

**Lemma 3.** *Suppose Assumption 2.2, and Assumption 3.2 hold. Then we have*

$$\mathbb{E}\left[\|\gamma_t - \gamma_*\|^k\right] = O\left(\sqrt{d_z^k \beta_t^{\,k}}\right) \quad for \quad k = 1, 2, 4. \tag{26}$$

We proceed by noting that if $\gamma_*$, $\Sigma_Z$, and $\Sigma_{ZY}$ were known beforehand, one could use the following deterministic gradient updates to obtain $\theta_*$.

$$\widetilde{\theta}_{t+1} = \widetilde{\theta}_t - \alpha_{t+1} \gamma_*^\top \left(\Sigma_Z \gamma_* \widetilde{\theta}_t - \Sigma_{ZY}\right). \tag{27}$$

**Lemma 4.** *Let Assumption 2.3 be true. Then, choosing $\eta_k = O(k^{-a})$ with $1/2 < a < 1$, we have $\|\widetilde{\theta}_t - \theta_*\| = O\left(\exp(-t^{1-a})\right).$*

Define the sequence $\delta_t := \theta_t - \widetilde{\theta}_t$. We will establish the convergence rate of $\mathbb{E}\left[\|\delta_t\|_2^2\right]$. From (13), and (27), we have the following expansion of $\delta_{t+1}$.

$$\delta_{t+1} = Q_t \delta_t + \alpha_{t+1} D_t \theta_t + \alpha_{t+1}(\gamma_t - \gamma_*)^\top \Sigma_{ZY} - \alpha_{t+1} \gamma_t^\top \xi_{Z_t Y_t} + \alpha_{t+1} \gamma_t^\top \xi_{Z_t} \gamma_t \theta_t, \tag{28}$$

where

$$Q_t := (I - \alpha_{t+1} \gamma_*^\top \Sigma_Z \gamma_*),$$
$$D_t := \gamma_*^\top \Sigma_Z \gamma_* - \gamma_t^\top \Sigma_Z \gamma_t.$$

First we will establish an intermediate bound on $\mathbb{E}\left[\|\delta_t\|^2\right]$. To do so, we will need the following result which shows that $\mathbb{E}\left[\|\theta_t - \theta_*\|_2^4\right]$ is bounded for all $t$ which we prove in Section E.2.

**Lemma 5 (Boundedness of fourth moment of $\|\theta_t - \theta_*\|$).** *Let the conditions in Theorem 2 be true. Then, choosing $\alpha_t, \beta_t$ such that $\alpha_t \leq d_z^{-4\varkappa - \vartheta/2}$, and $\sum_{t=1}^\infty (\alpha_t^2 + \alpha_t \sqrt{\beta_t}) < \infty$, we have $\mathbb{E}\left[\|\theta_t - \theta_*\|_2^4\right]$ is bounded by some constant $M > 0$.*

**Lemma 6 (Intermediate bound on $\mathbb{E}[\|\delta_t\|_2^2]$).** *Let the conditions in Theorem 2 be true. We have the following intermediate bound on $\mathbb{E}\left[\|\delta_t\|^2\right]$:*

$$\mathbb{E}\left[\|\delta_t\|^2\right] = O\left(\beta_t d_z^{1+2\varkappa} + \alpha_{t+1} d_z^{4\varkappa + \vartheta/2} + \sqrt{d_z \beta_t}\right). \tag{29}$$

*Proof of Lemma 6.* Recall the update for $\delta_{t+1}$ obtained in (28).

$$\delta_{t+1} = Q_t \delta_t + \alpha_{t+1} D_t \theta_t + \alpha_{t+1}(\gamma_t - \gamma_*)^\top \Sigma_{ZY} - \alpha_{t+1} \gamma_t^\top \xi_{Z_t Y_t} + \alpha_{t+1} \gamma_t^\top \xi_{Z_t} \gamma_t \theta_t.$$

Then,

$$\begin{aligned}
\|\delta_{t+1}\|_2^2 =& \delta_t^\top Q_t^2 \delta_t + \alpha_{t+1}^2 \|D_t \theta_t + (\gamma_t - \gamma_*)^\top \Sigma_{ZY} - \gamma_t^\top \xi_{Z_t Y_t} + \gamma_t^\top \xi_{Z_t} \gamma_t \theta_t\|^2 \\
&+ 2\alpha_{t+1} \delta_t^\top Q_t \left(D_t \theta_t + (\gamma_t - \gamma_*)^\top \Sigma_{ZY}\right) \\
&+ 2\alpha_{t+1} \delta_t^\top Q_t \left(\gamma_t^\top \xi_{Z_t} \gamma_t \theta_t - \gamma_t^\top \xi_{Z_t Y_t}\right).
\end{aligned} \tag{30}$$

Then, choosing $\alpha_1(\|\gamma_*\|_2\lambda_Z)^2 < 1$, using Young's inequality and Assumption 2.2, from (30) we get,

$$
\begin{aligned}
\mathbb{E}_t\left[\|\delta_{t+1}\|_2^2\right] \leq &(1-\alpha_{t+1}\mu)\|\delta_t\|^2 + 4\alpha_{t+1}^2\left(\|D_t\theta_t\|^2 + \|(\gamma_t-\gamma_*)^\top\Sigma_{ZY}\|^2\right) \\
&+ 4\alpha_{t+1}^2\left(\|\gamma_t\|_2^2\mathbb{E}\left[\|\xi_{Z_tY_t}\|_2^2\right] + \|\gamma_t\|_2^4\mathbb{E}\left[\|\xi_{Z_t}\|^2\right]\|\theta_t\|^2\right) \\
&+ 2\alpha_{t+1}\delta_t^\top Q_t\left(D_t\theta_t + (\gamma_t-\gamma_*)^\top\Sigma_{ZY}\right) \\
\lesssim &(1-\alpha_{t+1}\mu)\|\delta_t\|^2 + 4\alpha_{t+1}^2\left(\|D_t\|^2\|\theta_t\|^2 + \|(\gamma_t-\gamma_*)^\top\Sigma_{ZY}\|^2\right) \\
&+ 4C\alpha_{t+1}^2\left(d_z^{2\varkappa+\vartheta/2} + d_z^{4\varkappa+\vartheta/2}\|\theta_t\|^2\right) + \\
&2\alpha_{t+1}\delta_t^\top Q_t\left(D_t\theta_t + (\gamma_t-\gamma_*)^\top\Sigma_{ZY}\right),
\end{aligned}
$$

where the last inequality follows by Assumption 3.1, and Assumption 3.3.

Now, taking expectation on both sides, we obtain

$$
\begin{aligned}
\mathbb{E}\left[\|\delta_{t+1}\|_2^2\right] \lesssim &(1-\alpha_{t+1}\mu)\mathbb{E}\left[\|\delta_t\|^2\right] + 4\alpha_{t+1}^2\left(\mathbb{E}\left[\|D_t\|^2\|\theta_t\|^2\right] + \mathbb{E}\left[\|(\gamma_t-\gamma_*)^\top\Sigma_{ZY}\|^2\right]\right) \\
&+ 4C\alpha_{t+1}^2\left(d_z^{2\varkappa+\vartheta/2} + d_z^{4\varkappa+\vartheta/2}\mathbb{E}\left[\|\theta_t\|^2\right]\right) \\
&+ 2\alpha_{t+1}\left(\mathbb{E}\left[|\delta_t^\top Q_tD_t\theta_t|\right] + \mathbb{E}\left[|\delta_t^\top Q_t(\gamma_t-\gamma_*)^\top\Sigma_{ZY}|\right]\right).
\end{aligned}
\tag{31}
$$

Now, the following bounds are true:

1. We have that

$$
\alpha_{t+1}^2\mathbb{E}\left[\|D_t\|^2\|\theta_t\|^2\right] \leq \alpha_{t+1}^2\sqrt{\mathbb{E}\left[\|D_t\|_2^4\right]\mathbb{E}\left[\|\theta_t\|_2^4\right]} \lesssim d_z^{1+2\varkappa}\alpha_{t+1}^2\beta_t,
\tag{32}
$$

where the first inequality follows by Cauchy-Schwarz inequality, the second inequality follows by (42), and Lemma 5.

2. Using $\Sigma_{ZY} = O(1)$, and Lemma 3, we get

$$
\alpha_{t+1}^2\mathbb{E}\left[\|(\gamma_t-\gamma_*)^\top\Sigma_{ZY}\|^2\right] \lesssim d_z\beta_t\alpha_{t+1}^2.
\tag{33}
$$

3. We have that

$$
\begin{aligned}
\alpha_{t+1}\mathbb{E}\left[|\delta_t^\top Q_tD_t\theta_t|\right] \leq &\alpha_{t+1}\mathbb{E}\left[\|\delta_t\|_2\|Q_t\|_2\|D_t\|_2\|\theta_t\|_2\right] \\
\leq &\frac{\alpha_{t+1}\mu}{16}\mathbb{E}\left[\|\delta_t\|^2\right] + \frac{4\alpha_{t+1}}{\mu}\sqrt{\mathbb{E}\left[\|D_t\|_2^4\right]\mathbb{E}\left[\|\theta_t\|_2^4\right]} \\
\lesssim &\frac{\alpha_{t+1}\mu}{16}\mathbb{E}\left[\|\delta_t\|^2\right] + \frac{4d_z^{1+2\varkappa}\alpha_{t+1}\beta_t}{\mu},
\end{aligned}
\tag{34}
$$

where the first inequality follows by Hölder's inequality, the second inequality follows by Young's inequality, Cauchy-Schwarz inequality, and $\|Q_t\|_2 < 1$, and the third inequality follows by (42), and Lemma 5.

4. Using $\|Q_t\|_2 < 1$, $\|\Sigma_{ZY}\|_2 = O(1)$, Cauchy-Schwarz inequality, and Lemma 3, we get,

$$
\begin{aligned}
&\alpha_{t+1}\mathbb{E}\left[|\delta_t^\top Q_t(\gamma_t-\gamma_*)^\top\Sigma_{ZY}|\right] \\
\lesssim &\alpha_{t+1}\mathbb{E}\left[\|\delta_t\|_2\|\gamma_t-\gamma_*\|_2\right] \\
\leq &\alpha_{t+1}\sqrt{\mathbb{E}\left[\|\delta_t\|_2^2\right]\mathbb{E}\left[\|\gamma_t-\gamma_*\|_2^2\right]} \\
\leq &\frac{\sqrt{d_z\beta_t}\alpha_{t+1}}{2} + \frac{\sqrt{d_z\beta_t}\alpha_{t+1}\mathbb{E}\left[\|\delta_t\|_2^2\right]}{2}.
\end{aligned}
\tag{35}
$$

Combining (31), (32), (33), (34), (35), and Lemma 5, we have

$$
\mathbb{E}\left[\|\delta_{t+1}\|_2^2\right]
$$

$$
\begin{aligned}
&\lesssim (1 - \alpha_{t+1}\mu)\mathbb{E}\left[\|\delta_t\|^2\right] + 4\alpha_{t+1}^2\beta_t d_z^{1+2\varkappa} + 4C\alpha_{t+1}^2 d_z^{4\varkappa+\vartheta/2} \\
&\quad + 2\alpha_{t+1}\left(\mu\mathbb{E}\left[\|\delta_t\|^2\right]/16 + 4d_z^{1+2\varkappa}\beta_t/\mu + \sqrt{d_z\beta_t}/2 + \sqrt{d_z\beta_t}\mathbb{E}\left[\|\delta_t\|_2^2\right]/2\right) \quad (36) \\
&\lesssim (1 - 7\mu\alpha_{t+1}/8 + \alpha_{t+1}\sqrt{d_z\beta_t})\mathbb{E}\left[\|\delta_t\|^2\right] + (8\alpha_{t+1}\beta_t d_z^{1+2\varkappa}/\mu + 4C\alpha_{t+1}^2 d_z^{4\varkappa+\vartheta/2}) \\
&\quad + \alpha_{t+1}\sqrt{d_z\beta_t} \\
&\lesssim (1 - 3\mu\alpha_{t+1}/4)\mathbb{E}\left[\|\delta_t\|^2\right] + (8\alpha_{t+1}\beta_t d_z^{1+2\varkappa}/\mu + 4C\alpha_{t+1}^2 d_z^{4\varkappa+\vartheta/2}) + \alpha_{t+1}\sqrt{d_z\beta_t}. \quad (37)
\end{aligned}
$$

In the above, the third inequality follows by choosing $\beta_t \leq \mu^2/(64d_z)$, and $\alpha_{t+1}\sqrt{d_z\beta_t} < 1$. Then, from (37), we have

$$
\mathbb{E}\left[\|\delta_t\|_2^2\right] = O\left(\beta_t d_z^{1+2\varkappa} + \alpha_{t+1}d_z^{4\varkappa+\vartheta/2} + \sqrt{d_z\beta_t}\right).
$$

$\square$

Coming back to the proof of Theorem 2, observe that, we can sharpen the bound in (35) using Lemma 6 which allows us to avoid the use of Young's inequality. This leads to the following improved version of the recursion in (37) using which we can improve the term $\sqrt{d_z\beta_t}$ in (29) as follows:

$$
\begin{aligned}
\mathbb{E}\left[\|\delta_{t+1}\|_2^2\right] &\lesssim (1 - 7\mu\alpha_{t+1}/8)\mathbb{E}\left[\|\delta_t\|^2\right] \\
&\quad + \alpha_{t+1}O\left(\beta_t d_z^{1+2\varkappa} + \alpha_{t+1}d_z^{4\varkappa+\vartheta/2} + \sqrt{\alpha_{t+1}\beta_t}d_z^{1/2+2\varkappa+\vartheta/4} + (\beta_t d_z)^{3/4}\right) \\
&= O\left(\beta_t d_z^{1+2\varkappa} + \alpha_{t+1}d_z^{4\varkappa+\vartheta/2} + \sqrt{\alpha_{t+1}\beta_t}d_z^{1/2+2\varkappa+\vartheta/4} + (\beta_t d_z)^{3/4}\right).
\end{aligned}
$$

In fact, this trick can be used repeatedly to sharpen the bound even further as shown in Lemma 7.

**Lemma 7** (**Final improved bound on** $\mathbb{E}[\|\delta_t\|_2^2]$). *Let the conditions in Theorem 2 be true. Then using Lemma 6, we have,*

$$
\begin{aligned}
&\mathbb{E}\left[\|\delta_{t+1}\|_2^2\right] \\
&\lesssim O\left((d_z\beta_t)^{1-2^{-r-1}} + \sum_{i=0}^{r}\left(\alpha_{t+1}^{2^{-i}}\beta_t^{1-2^{-i}}d_z^{1+(4\varkappa+\vartheta/2-1)2^{-i}} + \beta_t(1+\alpha_{t+1}^{2^{-i}})d_z^{1+2^{1-i}\varkappa}\right)\right),
\end{aligned}
$$

*where $r$ is any non-negative integer.*

*Proof of Lemma 7.* If we have

$$
\mathbb{E}\left[\|\delta_t\|^2\right] = O\left(\alpha_{t+1}d_z^{4\varkappa+\vartheta/2} + \beta_t d_z^{1+2\varkappa} + \sqrt{d_z\beta_t}\right),
$$

then from (35), we have,

$$
\begin{aligned}
\mathbb{E}\left[|\delta_t^\top Q_t(\gamma_t - \gamma_*)^\top \Sigma_{ZY}|\right] &\lesssim \sqrt{\mathbb{E}\left[\|\delta_t\|_2^2\right]\mathbb{E}\left[\|\gamma_t - \gamma_*\|_2^2\right]} \\
&= O\left(\sqrt{\alpha_{t+1}\beta_t}d_z^{1/2+2\varkappa+\vartheta/4} + \beta_t d_z^{1+\varkappa} + (d_z\beta_t)^{3/4}\right). \quad (38)
\end{aligned}
$$

Then, similar to (36), we have,

$$
\begin{aligned}
&\mathbb{E}\left[\|\delta_{t+1}\|_2^2\right] \\
&\lesssim (1-\alpha_{t+1}\mu)\mathbb{E}\left[\|\delta_t\|^2\right] + 4\alpha_{t+1}^2\beta_t d_z^{1+2\varkappa} + 4C\alpha_{t+1}^2 d_z^{4\varkappa+\vartheta/2} \\
&\quad + 2\alpha_{t+1}\left(\mu\mathbb{E}\left[\|\delta_t\|^2\right]/16 + 4d_z^{1+2\varkappa}\beta_t/\mu + \sqrt{\alpha_{t+1}\beta_t}d_z^{1/2+2\varkappa+\vartheta/4} + \beta_t d_z^{1+\varkappa} + (d_z\beta_t)^{3/4}\right) \\
&\lesssim (1 - 7\mu\alpha_{t+1}/8)\mathbb{E}\left[\|\delta_t\|^2\right] \\
&\quad + \alpha_{t+1}O\left((d_z\beta_t)^{3/4} + \sum_{i=0}^{1}\left(\alpha_{t+1}^{2^{-i}}\beta_t^{1-2^{-i}}d_z^{1+(4\varkappa+\vartheta/2-1)2^{-i}} + \beta_t(1+\alpha_{t+1}^{2^{-i}})d_z^{1+2^{1-i}\varkappa}\right)\right) \\
&= O\left((d_z\beta_t)^{3/4} + \sum_{i=0}^{1}\left(\alpha_{t+1}^{2^{-i}}\beta_t^{1-2^{-i}}d_z^{1+(4\varkappa+\vartheta/2-1)2^{-i}} + \beta_t(1+\alpha_{t+1}^{2^{-i}})d_z^{1+2^{1-i}\varkappa}\right)\right).
\end{aligned}
$$

Now if we repeat this step $r$ number of times (where $r$ is to be set later), by progressive sharpening we get the following bound.

$$\mathbb{E}\left[\|\delta_{t+1}\|_2^2\right]$$

$$\lesssim O\left((d_z\beta_t)^{1-2^{-r-1}} + \sum_{i=0}^{r}\left(\alpha_{t+1}^{2^{-i}}\beta_t^{1-2^{-i}}d_z^{1+(4\varkappa+\vartheta/2-1)2^{-i}} + \beta_t(1+\alpha_{t+1}^{2^{-i}})d_z^{1+2^{1-i}\varkappa}\right)\right).$$

$\square$

Coming back to the proof of Theorem 2, we have that by combining Lemma 4, and Lemma 7,

$$\mathbb{E}\left[\|\theta_t - \theta_*\|^2\right] \leq 2\mathbb{E}\left[\|\delta_t\|^2\right] + 2\mathbb{E}\left[\|\widetilde{\theta}_t - \theta_*\|^2\right]$$

$$=O\left((d_z\beta_t)^{1-2^{-r-1}} + \sum_{i=0}^{r}\left(\alpha_{t+1}^{2^{-i}}\beta_t^{1-2^{-i}}d_z^{1+(4\varkappa+\vartheta/2-1)2^{-i}} + \beta_t(1+\alpha_{t+1}^{2^{-i}})d_z^{1+2^{1-i}\varkappa}\right)\right). \quad (39)$$

Now, in (39), for some arbitrarily small number $\iota > 0$, choosing

$$\alpha_t = \min(0.5d_z^{-4\varkappa-\vartheta/2}\lambda_Z^{-1}C_\gamma^{-2}, 0.5(\|\gamma_*\|_2\lambda_Z)^{-2})t^{-1+\iota/2}, \qquad \beta_t = \mu^2 d_z^{-1-2\varkappa}t^{-1+\iota/2}/128,$$

and setting $r = \lceil\log_2\left((\iota/2)^{-1} - 1\right) - 1\rceil$ we get,

$$\mathbb{E}\left[\|\theta_t - \theta_*\|^2\right] = O\left(\max\left(t^{-1+\iota}, t^{-1+\iota/2}\log((\iota/2)^{-1} - 1)\right)\right).$$

$\square$

## E.2 Proof of Lemma 5

*Proof.* Using the form of $\theta_*$, from (13) we get,

$$\theta_{t+1} - \theta_* = \widehat{Q}_t(\theta_t - \theta_*) + \alpha_{t+1}(\gamma_t - \gamma_*)^\top\Sigma_{ZY} + \alpha_{t+1}D_t\theta_* + \alpha_{t+1}\gamma_t^\top\xi_{Z_t}\gamma_t(\theta_t - \theta_*)$$

$$+ \alpha_{t+1}\gamma_t^\top\xi_{Z_t}\gamma_t\theta_* + \alpha_{t+1}\gamma_t^\top\xi_{Z_tY_t}. \quad (40)$$

where $\widehat{Q}_t := \left(I - \alpha_{t+1}\gamma_t^\top\Sigma_Z\gamma_t\right) = Q_t + \alpha_{t+1}D_t$. Recall that $D_t = \gamma_*^\top\Sigma_Z\gamma_* - \gamma_t^\top\Sigma_Z\gamma_t$. By Assumption 3.3, we have the following bound on $\|D_t\|_2$.

$$\|D_t\|_2 = O(\lambda_Z C_\gamma^2 d_z^{2\varkappa}). \quad (41)$$

We have the following bound on $\mathbb{E}\left[\|D_t\|_2^4\right]$ by Lemma 3.

$$\mathbb{E}\left[\|D_t\|_2^4\right] = \mathbb{E}\left[\|(\gamma_* - \gamma_t)^\top\Sigma_Z\gamma_* + \gamma_t^\top\Sigma_Z(\gamma_* - \gamma_t)\|_2^4\right] = O(d_z^{2+4\varkappa}\beta_t^2). \quad (42)$$

From (40), we have

$$\|\theta_{t+1} - \theta_*\|_2^2 \leq (\theta_t - \theta_*)^\top\widehat{Q}_t^2(\theta_t - \theta_*) + 3\alpha_{t+1}^2\|\gamma_t^\top\xi_{Z_t}\gamma_t(\theta_t - \theta_*)\|_2^2$$

$$+ 2\alpha_{t+1}(\theta_t - \theta_*)^\top\widehat{Q}_t(\gamma_t - \gamma_*)^\top\Sigma_{ZY}$$

$$+ 2\alpha_{t+1}(\theta_t - \theta_*)^\top\widehat{Q}_tD_t\theta_* + A_{1,t} + A_{2,t}, \quad (43)$$

where

$$A_{1,t} = \alpha_{t+1}^2\left(\|(\gamma_t - \gamma_*)^\top\Sigma_{ZY}\|_2^2 + \|D_t\theta_*\|_2^2\right.$$

$$\left. + 2\Sigma_{ZY}^\top(\gamma_t - \gamma_*)D_t\theta_* + 3\|\gamma_t^\top\xi_{Z_t}\gamma_t\theta_*\|_2^2 + 3\|\gamma_t^\top\xi_{Z_tY_t}\|_2^2\right), \quad (44)$$

and

$$A_{2,t} = 2\alpha_{t+1}(\widehat{Q}_t(\theta_t - \theta_*) + \alpha_{t+1}(\gamma_t - \gamma_*)^\top\Sigma_{ZY}$$

$$+ \alpha_{t+1}D_t\theta_*)^\top(\gamma_t^\top\xi_{Z_t}\gamma_t(\theta_t - \theta_*) + \gamma_t^\top\xi_{Z_t}\gamma_t\theta_* + \gamma_t^\top\xi_{Z_tY_t}).$$

Define

$$A_{3,t} := 3\alpha_{t+1}^2\|\gamma_t^\top\xi_{Z_t}\gamma_t(\theta_t - \theta_*)\|_2^2 + 2\alpha_{t+1}(\theta_t - \theta_*)^\top\widehat{Q}_t(\gamma_t - \gamma_*)^\top\Sigma_{ZY}$$

$$+ 2\alpha_{t+1}(\theta_t - \theta_*)^\top\widehat{Q}_tD_t\theta_* + A_{1,t} + A_{2,t}. \quad (45)$$

Then, choosing $C_\gamma^2 d_z^{2\varkappa}\lambda_Z\alpha_{t+1} < 1$, which ensures $\|\widehat{Q}_t\| \leq 1$, we have

$$\|\theta_{t+1} - \theta_*\|_2^4 \leq \|\theta_t - \theta_*\|_2^4 + 2(\theta_t - \theta_*)^\top\widehat{Q}_t^2(\theta_t - \theta_*)A_{3,t} + A_{3,t}^2. \quad (46)$$

We now have the following bounds:

1. Using Assumption 3.1, and Assumption 3.3,

$$\alpha_{t+1}^4 \mathbb{E}\left[\|\gamma_t^{\top}\xi_{Z_t}\gamma_t(\theta_t - \theta_*)\|_2^4\right] \lesssim d_z^{8\varkappa+\vartheta}\alpha_{t+1}^4\mathbb{E}\left[\|\theta_t - \theta_*\|_2^4\right].\tag{47}$$

2. We have that

$$\mathbb{E}\left[((\theta_t - \theta_*)^{\top}\widehat{Q}_t(\gamma_t - \gamma_*)^{\top}\Sigma_{ZY})^2\right]$$
$$\lesssim \mathbb{E}\left[\|\theta_t - \theta_*\|^2\|\gamma_t - \gamma_*\|^2\right]$$
$$\leq \sqrt{\mathbb{E}\left[\|\theta_t - \theta_*\|_2^4\right]\mathbb{E}\left[\|\gamma_t - \gamma_*\|_2^4\right]}$$
$$\leq d_z\beta_t\left(1 + \mathbb{E}\left[\|\theta_t - \theta_*\|_2^4\right]\right)/2,\tag{48}$$

where, the first inequality follows by $\|\widehat{Q}_t\|_2 = O(1)$, and $\|\Sigma_{ZY}\|_2 = O(1)$. The second inequality follows by Cauchy-Schwarz inequality. The last inequality follows by $\sqrt{ab} \leq (a + b)/2$, and Lemma 3.

3. We have that

$$\mathbb{E}\left[((\theta_t - \theta_*)^{\top}\widehat{Q}_t D_t\theta_*)^2\right]$$
$$\lesssim \mathbb{E}\left[\|\theta_t - \theta_*\|_2^2\|D_t\|_2^2\right]$$
$$\leq \sqrt{\mathbb{E}\left[\|\theta_t - \theta_*\|_2^4\right]\mathbb{E}\left[\|D_t\|_2^4\right]}$$
$$\lesssim d_z^{1+2\varkappa}\beta_t\left(1 + \mathbb{E}\left[\|\theta_t - \theta_*\|_2^4\right]\right)/2,\tag{49}$$

where, the first inequality follows by $\|\widehat{Q}_t\|_2 = O(1)$, and $\|\theta_*\|_2 = O(1)$. The second inequality follows by Cauchy-Schwarz inequality. The last inequality follows by $\sqrt{ab} \leq (a + b)/2$, and (42).

4. Using Assumption 3.1, Assumption 3.3, (42), and Lemma 3, we have

$$\mathbb{E}\left[A_{1,t}^2\right] = O\left(d_z^{8\varkappa+\vartheta}\alpha_{t+1}^4\right).\tag{50}$$

.

5. Using Young's inequality, Assumption 3.1, Assumption 3.3, Lemma 3, $\|\Sigma_{ZY}\|_2 = O(1)$, $\|\theta_*\|_2 = O(1)$, and (42), we have

$$\mathbb{E}\left[A_{2,t}^2\right] \leq 2\alpha_{t+1}^2\mathbb{E}\left[\|\widehat{Q}_t(\theta_t - \theta_*) + \alpha_{t+1}(\gamma_t - \gamma_*)^{\top}\Sigma_{ZY} + \alpha_{t+1}D_t\theta_*\|_2^4\right]$$
$$+ 2\alpha_{t+1}^2\mathbb{E}\left[\|\gamma_t^{\top}\xi_{Z_t}\gamma_t(\theta_t - \theta_*) + \gamma_t^{\top}\xi_{Z_t}\gamma_t\theta_* + \gamma_t^{\top}\xi_{Z_tY_t}\|_2^4\right]$$
$$\lesssim \alpha_{t+1}^2 d_z^{8\varkappa+\vartheta}(1 + \mathbb{E}\left[\|\theta_t - \theta_*\|_2^4\right]).\tag{51}$$

6. Using $\|\widehat{Q}_t\|_2 = O(1)$, Assumption 3.1, and Assumption 3.3,

$$\alpha_{t+1}^2\mathbb{E}\left[(\theta_t - \theta_*)^{\top}\widehat{Q}_t^2(\theta_t - \theta_*)\|\gamma_t^{\top}\xi_{Z_t}\gamma_t(\theta_t - \theta_*)\|_2^2\right] \lesssim \alpha_{t+1}^2 d_z^{4\varkappa+\vartheta/2}\mathbb{E}\left[\|\theta_t - \theta_*\|_2^4\right].\tag{52}$$

7. We have that

$$\alpha_{t+1}\mathbb{E}\left[|(\theta_t - \theta_*)^{\top}\widehat{Q}_t^2(\theta_t - \theta_*)(\theta_t - \theta_*)^{\top}\widehat{Q}_t(\gamma_t - \gamma_*)^{\top}\Sigma_{ZY}|\right]$$
$$\lesssim \alpha_{t+1}\mathbb{E}\left[\|\theta_t - \theta_*\|_2^3\|\gamma_t - \gamma_*\|_2\right]$$
$$\leq \alpha_{t+1}\left(\mathbb{E}\left[\|\theta_t - \theta_*\|_2^4\right]\right)^{3/4}\left(\mathbb{E}\left[\|\gamma_t - \gamma_*\|_2^4\right]\right)^{1/4}$$
$$\leq \alpha_{t+1}\sqrt{d_z\beta_t}\left(\mathbb{E}\left[\|\theta_t - \theta_*\|_2^4\right]\right)^{3/4}$$
$$\leq \frac{3\alpha_{t+1}\sqrt{d_z\beta_t}}{4}\mathbb{E}\left[\|\theta_t - \theta_*\|_2^4\right] + \frac{\alpha_{t+1}\sqrt{d_z\beta_t}}{4},\tag{53}$$

where, the first inequality follows by $\|\widehat{Q}_t\|_2 = O(1)$, and $\|\Sigma_{ZY}\|_2 = O(1)$, the second inequality follows by Cauchy-Schwarz inequality, the third inequality follows by Lemma 3 and the fourth inequality follows by Young's inequality.

8. Similar to (53), we have,

$$\alpha_{t+1}\mathbb{E}\left[|(\theta_t - \theta_*)^\top \widehat{Q}_t^2(\theta_t - \theta_*)(\theta_t - \theta_*)^\top \widehat{Q}_t D_t \theta_*|\right]$$

$$\leq \frac{3d_z^{1/2+\varkappa}\alpha_{t+1}\sqrt{\beta_t}}{4}\mathbb{E}\left[\|\theta_t - \theta_*\|_2^4\right] + \frac{d_z^{1/2+\varkappa}\alpha_{t+1}\sqrt{\beta_t}}{4}. \tag{54}$$

9. Using $\|\widehat{Q}_t\|_2 = O(1)$, Cauchy-Schwarz inequality, (50), and Young's inequality,

$$\mathbb{E}\left[(\theta_t - \theta_*)^\top \widehat{Q}_t^2(\theta_t - \theta_*)A_{1,t}\right]$$

$$\leq \mathbb{E}\left[\|\theta_t - \theta_*\|_2^2 A_{1,t}\right]$$

$$\leq \sqrt{\mathbb{E}\left[\|\theta_t - \theta_*\|_2^4\right]\mathbb{E}\left[A_{1,t}^2\right]}$$

$$\lesssim d_z^{4\varkappa+\vartheta/2}\alpha_{t+1}^2\left(1 + \mathbb{E}\left[\|\theta_t - \theta_*\|_2^4\right]\right). \tag{55}$$

10. By Assumption 2.2, we have,

$$\mathbb{E}_t\left[(\theta_t - \theta_*)^\top \widehat{Q}_t^2(\theta_t - \theta_*)A_{2,t}\right] = 0. \tag{56}$$

Now using Jensen's inequality, and combining (47), (48), (49), (50), and (51), we have,

$$\mathbb{E}\left[A_{3,t}^2\right] \leq 45\alpha_{t+1}^4\mathbb{E}\left[\|\gamma_t^\top \xi_{Z_t}\gamma_t(\theta_t - \theta_*)\|_2^4\right] + 20\alpha_{t+1}^2\mathbb{E}\left[((\theta_t - \theta_*)^\top \widehat{Q}_t(\gamma_t - \gamma_*)^\top \Sigma_{ZY})^2\right]$$

$$+ 20\alpha_{t+1}^2\mathbb{E}\left[((\theta_t - \theta_*)^\top \widehat{Q}_t D_t \theta_*)^2\right] + 5\mathbb{E}\left[A_{1,t}^2\right] + 5\mathbb{E}\left[A_{2,t}^2\right]$$

$$\lesssim \alpha_{t+1}^4 d_z^{\vartheta_7+8\varkappa}\mathbb{E}\left[\|\theta_t - \theta^*\|_2^4\right] + d_z\alpha_{t+1}^2\beta_t\left(1 + \mathbb{E}\left[\|\theta_t - \theta_*\|_2^4\right]\right)$$

$$+ \alpha_{t+1}^2 d_z^{1+2\varkappa}\beta_t\left(1 + \mathbb{E}\left[\|\theta_t - \theta_*\|_2^4\right]\right) + d_z^{8\varkappa+\vartheta}\alpha_{t+1}^4 + \alpha_{t+1}^2 d_z^{8\varkappa+\vartheta}\left(1 + \mathbb{E}\left[\|\theta_t - \theta_*\|_2^4\right]\right)$$

$$\lesssim \alpha_{t+1}^2 d_z^{8\varkappa+\vartheta}\left(1 + \mathbb{E}\left[\|\theta_t - \theta_*\|_2^4\right]\right). \tag{57}$$

Combining (52), (53), (54), (55), and (56), we get,

$$\mathbb{E}\left[(\theta_t - \theta_*)^\top \widehat{Q}_t^2(\theta_t - \theta_*)A_{3,t}\right]$$

$$\lesssim \alpha_{t+1}^2 d_z^{4\varkappa+\vartheta/2}\mathbb{E}\left[\|\theta_t - \theta_*\|_2^4\right] + \frac{3\alpha_{t+1}\sqrt{d_z\beta_t}}{4}\mathbb{E}\left[\|\theta_t - \theta_*\|_2^4\right] + \frac{\alpha_{t+1}\sqrt{d_z\beta_t}}{4}$$

$$+ \frac{3d_z^{1/2+\varkappa}\alpha_{t+1}\sqrt{\beta_t}}{4}\mathbb{E}\left[\|\theta_t - \theta_*\|_2^4\right] + \frac{d_z^{1/2+\varkappa}\alpha_{t+1}\sqrt{\beta_t}}{4} + d_z^{4\varkappa+\vartheta/2}\alpha_{t+1}^2\left(1 + \mathbb{E}\left[\|\theta_t - \theta_*\|_2^4\right]\right)$$

$$\lesssim (\alpha_{t+1}^2 d_z^{4\varkappa+\vartheta/2} + \alpha_{t+1}\sqrt{\beta_{t+1}}d_z^{1/2+\varkappa})(1 + \|\theta_t - \theta^*\|_2^4). \tag{58}$$

Combining (46), (57), and (58), we have,

$$\mathbb{E}\left[\|\theta_{t+1} - \theta_*\|_2^4\right] \lesssim (1 + \alpha_{t+1}^2 d_z^{8\varkappa+\vartheta} + \alpha_{t+1}\sqrt{\beta_{t+1}}d_z^{1/2+\varkappa})\left(1 + \mathbb{E}\left[\|\theta_t - \theta_*\|_2^4\right]\right). \tag{59}$$

Now choosing $\alpha_t, \beta_t$ such that $\alpha_t \leq d_z^{-4\varkappa-\vartheta/2}$, and $\sum_{t=1}^\infty (\alpha_{t+1}^2 + \alpha_{t+1}\sqrt{\beta_{t+1}}) < \infty$, we get

$$\mathbb{E}\left[\|\theta_t - \theta_*\|_2^4\right] \leq M, \tag{60}$$

for some constant $0 \leq M < \infty$. $\qquad\square$

### E.3 Comment on the convergence of (11)

We now discuss the convergence properties of the update sequence (11), which we refer to as the *conditional stochastic optimization* (CSO) based updates, which we restate below:

$$\theta_{t+1} = \theta_t - \alpha_{t+1}\gamma_t^\top Z_t(X_t^\top \theta_t - Y_t), \qquad \gamma_{t+1} = \gamma_t - \beta_{t+1}Z_t(Z_t^\top \gamma_t - X_t^\top).$$

Similar to (40), for the above updates, we have the following expansion:

$$\theta_{t+1} - \theta_* = \widehat{Q}_t(\theta_t - \theta_*) + \alpha_{t+1}(\gamma_t - \gamma_*)^\top \Sigma_{ZY} + \alpha_{t+1}D_t\theta_* + \alpha_{t+1}\gamma_t^\top \xi_{Z_t}\gamma_*(\theta_t - \theta_*)$$

$$+ \alpha_{t+1}\gamma_t{}^\top \xi_{Z_t}\gamma_*\theta_* + \alpha_{t+1}\gamma_t{}^\top \xi_{Z_tY_t} - \alpha_{t+1}\gamma_t{}^\top Z_t\epsilon_{2,t}{}^\top\theta_t,$$

where $\xi_{Z_t} = \Sigma_Z - Z_tZ_t^\top$, $\xi_{Z_tY_t} = \Sigma_{ZY} - Z_tY_t$, $\widehat{Q}_t := \left(I - \alpha_{t+1}\gamma_t{}^\top\Sigma_Z\gamma_*\right) = Q_t + \alpha_{t+1}D_t$, and $D_t = (\gamma_* - \gamma_t)\Sigma_Z\gamma_*$.

Recall that the reason for the initial divergence of the updates in (11) are the potential negative eigenvalues of $\gamma_t{}^\top\Sigma_Z\gamma_*$. Here we will show that if $\gamma_t{}^\top\Sigma_Z\gamma_*$ is positive semi-definite or $\gamma_t$ is close enough to $\gamma_*$ such that the negative eigenvalues (if any) are not too large in absolute values, then the updates in (11) indeed exhibit the same convergence rate as Algorithm 2.

**Assumption E.1.** *Let either of the following two conditions be true. For all $t \geq t_0$,*

1. *$\gamma_t\Sigma_Z\gamma_*$ is positive semidefinite.*
2. *$\|\gamma_t - \gamma_*\|^2 \lesssim d_z\beta_t$.*

Note that Condition 1 of Assumption E.1 is an idealized condition which is difficult to ensure for all $t$ in reality. But of course if this is true, then $\gamma_t\Sigma_Z\gamma_*$ does not have a negative eigenvalue to cause divergence and the proof then follows exactly like Lemma 5.

Hence, we will focus on the more realistic Condition 2 of Assumption E.1 which holds true almost surely [PJ92]. Since we are interested in the asymptotic rate of convergence of CSO updates (due to the requirement of Assumption E.1), we will only concentrate on the iterations $t \geq t_0$. In this case, the proof steps are similar to Theorem 2 except for two major differences, that we discuss below.

### Difference 1: Potential negative definiteness of $\gamma_t{}^\top\Sigma_Z\gamma_*$:

Under Condition 2, $\gamma_t{}^\top\Sigma_Z\gamma_*$ can indeed be negative definite. In general, if $\gamma_t{}^\top\Sigma_Z\gamma_*$ is negative definite then that is undesirable as we explain Section 3. In terms of the proof, we can no longer write $(\theta_t - \theta^*)^\top\widehat{Q}_t^\top\widehat{Q}_t(\theta_t - \theta^*) \leq \|\theta_t - \theta_*\|^2$ (which was possible to do in (43) in the proof of Lemma 5). Subsequently, (46) breaks down. But we will show that under Condition 2 the negative eigenvalues are not too large in terms of absolute values. Specifically, we can write,

$$
\begin{aligned}
&(\theta_t - \theta^*)^\top\widehat{Q}_t^\top\widehat{Q}_t(\theta_t - \theta^*) \\
=&(\theta_t - \theta^*)^\top(Q_t^2 + \alpha_{t+1}Q_t^\top D_t + \alpha_{t+1}D_t^\top Q_t + \alpha_{t+1}^2 D_t^\top D_t)(\theta_t - \theta^*) \\
\leq&(1 + 2\alpha_{t+1}\|D_t\|)\|\theta_t - \theta_*\|^2 + \alpha_{t+1}^2\|D_t\|^2\|\theta_t - \theta_*\|^2 \\
\leq&(1 + 2\alpha_{t+1}\sqrt{d_z\beta_t})\|\theta_t - \theta_*\|^2 + \alpha_{t+1}^2\|D_t\|^2\|\theta_t - \theta_*\|^2.
\end{aligned}
\tag{61}
$$

The term $\alpha_{t+1}^2\|D_t\|^2\|\theta_t - \theta_*\|^2$ is of the order of $A_{3,t}$ defined in (45). Now $\alpha_{t+1}\sqrt{d_z\beta_t}$ is small enough in the sense that we choose the stepsizes such that $\sum_{t=1}^\infty(\alpha_{t+1}^2 + \alpha_{t+1}\sqrt{\beta_t}) < \infty$. Using this one can now show a similar bound as (59) and consequently show $\mathbb{E}\left[\|\theta_t - \theta_*\|^4\right]$ is bounded.

Now let us see what happens in the absence of Condition 2. Here one could use the fact $(1 + 2\alpha_{t+1}\|D_t\|) \lesssim (1 + 2C_\gamma\alpha_{t+1}d_z^\varkappa)$ which is too big. Recall that we want something at least of the order of $\alpha_{t+1}\sqrt{\beta_t}$ to show that $\theta_t$ sequence is bounded. One could also try to use the fact that $\mathbb{E}\left[\|D_t\|\right]$ is small by Lemma 3. But since $D_t$ and $\theta_t$ are interdependent, one needs to decouple them. One way to do this would be to use Cauchy-Shwarz inequalityas shown below.

$$\mathbb{E}\left[\|D_t\|\|\theta_t - \theta_*\|^2\right] \leq \sqrt{\mathbb{E}\left[\|D_t\|^2\right]\mathbb{E}\left[\|\theta_t - \theta_*\|^4\right]} \lesssim \sqrt{d_z\beta_t\mathbb{E}\left[\|\theta_t - \theta_*\|^4\right]}.$$

But that leads to the presence of $\mathbb{E}\left[\|\theta_t - \theta_*\|^4\right]$ in (43) which is potentially problematic due to the fact that on the left-hand side we have $\mathbb{E}\left[\|\theta_{t+1} - \theta_*\|^2\right]$.

### Difference 2: Presence of additional error term $\alpha_{t+1}\gamma_t{}^\top Z_t\epsilon_{2,t}{}^\top\theta_t$:

When comparing (12) with (40), yet another crucial difference is the presence of the term $\alpha_{t+1}\gamma_t{}^\top Z_t\epsilon_{2,t}{}^\top\theta_t$. We will show by the following observations that this error term gets absorbed by other terms already present in (40) without affecting the convergence rate. Specifically, the following holds.

1. Using the independence between $Z$, and $\epsilon_{2,t}$, and by Assumption 2.2, we have,

$$\mathbb{E}_t[(\widehat{Q}_t(\theta_t - \theta_*) + \alpha_{t+1}(\gamma_t - \gamma_*)^\top \Sigma_{ZY} + \alpha_{t+1} D_t \theta_* + \alpha_{t+1} \gamma_t^\top \xi_{Z_t} \gamma_* (\theta_t - \theta_*)$$
$$+ \alpha_{t+1} \gamma_t^\top \xi_{Z_t} \gamma_* \theta_*)^\top \gamma_t^\top Z_t \epsilon_{2,t}^\top \theta_t] = 0.$$

2. We also have that

$$\alpha_{t+1}^2 \mathbb{E}_t \left[ (\gamma_t^\top \xi_{Z_t Y_t})^\top \gamma_t^\top Z_t \epsilon_{2,t}^\top \theta_t \right]$$
$$= \alpha_{t+1}^2 (\gamma_t^\top \Sigma_Z \gamma_t \|\theta_*\|^2 + \gamma_t^\top \Sigma_Z \gamma_t \theta_*^\top (\theta_t - \theta_*))$$
$$\leq \alpha_{t+1}^2 (\gamma_t^\top \Sigma_Z \gamma_t \|\theta_*\|^2 + \|\gamma_t^\top \Sigma_Z \gamma_t (\theta_t - \theta_*)\|^2 + \|\theta_*\|^2)$$

This shows that the above term is of the same order as $A_{1,t}$ and $A_{3,t}$ defined in (44), and (45).

3. Finally, we have

$$\alpha_{t+1}^2 \mathbb{E}_t \left[ \|\gamma_t^\top Z_t \epsilon_{2,t}^\top \theta_t\|^2 \right] \lesssim \alpha_{t+1}^2 (\|\gamma_t\|^2 \|\theta_t - \theta_*\|^2 + \|\gamma_t\|^2 \|\theta_*\|^2).$$

So this term is of the order of $A_{3,t}$ as well.

Combining the above facts and following similar procedure as the proof of Theorem 2, one can show that the CSO updates achieve a similar rate under additional Assumption E.1.

