# OpenReview forum: "Stochastic Optimization Algorithms for Instrumental Variable Regression with Streaming Data"
_NeurIPS.cc/2024/Conference — NeurIPS 2024 poster_

### Official Review · Reviewer_AcA7 · 2024-07-03

**Soundness:** 3
**Presentation:** 4
**Contribution:** 2
**Rating:** 6
**Confidence:** 2

**Summary:**

This paper proposes two stochastic optimization algorithms for instrumental variable regression (IVaR) that operate on streaming data without requiring matrix inversions or mini-batches. When the true model is linear, the paper proves that TOSG-IVaR converges at a rate of $\mathcal{O}(\log T/T)$ and OTSG-IVaR at a rate of $O(1/T^{1 - \iota})$ for any $\iota > 0$, where $T$ is the number of iterations. The proposed approaches avoid the "forbidden regression" problem of having to estimate the nuisance parameter relating $Z$ and $X$. Numerical experiments validate the theoretical results.

**Strengths:**

1. The paper provides a novel perspective on instrumental variable regression (IVaR) by formulating it as a conditional stochastic optimization problem, which allows for the development of fully online algorithms that operate on streaming data.

2. The proposed algorithms, TOSG-IVaR and OTSG-IVaR, avoid the need for matrix inversions and mini-batches, making them computationally efficient and suitable for large-scale datasets.

3. By directly solving the conditional stochastic optimization problem, the algorithms avoid the "forbidden regression" issue and the need to approximate a dual variable over a continuous functional space, which is a limitation of prior minimax formulations.

4. This paper is well-structured. The background, contributions, and limitations are explicitly presented. The proof techniques is clearly summarized and the experimental results do support the performance of both algorithms.

**Weaknesses:**

1. The paper does not provide a comprehensive comparison with existing IVaR methods, particularly in terms of computational complexity and empirical performance on real-world datasets. Such a comparison would strengthen the paper's contributions.

2. The assumptions made for the theoretical analysis, such as the identifiability conditions and moment assumptions, may be restrictive in some practical settings. A discussion on the robustness of the proposed algorithms under violations of these assumptions would be valuable.

**Questions:**

The theoretical analysis focuses on the linear model setting. What are the challenges in extending the analysis to non-linear models, and what kind of assumptions or modifications would be required?

**Limitations:**

The limitations and future works are thoroughly discussed in the paper. Besides from that, I do not see more limitations.

---

> ### Author Rebuttal · Authors · 2024-08-07
>
> Dear Reviewer AcA7,
>
> Thank you for your insightful review. Here we provide response to your questions and concerns.
> ### **Weaknesses**
> **W1** - Please note that many existing IV methods are not applicable for online/streaming setting. We also wish to clarify that in Appendix B Table 1 we have compared the computational complexities (specifically, per-iteration arithmetic and memory complexity) of our method to that form [DVB23b] (which is a recently proposed online/streaming IV method) and demonstrated our benefits. In our experimental results (Section 4), we have also compared our algorithms against that in [DVB23b] and demonstrated the benefits.  In particular, we wish to highlight that our algorithm converges much faster.
>
> Finally, as a part of the rebuttal, we have provided empirical results on real-world dataset as well. Compared to [DVB23b], our method also performs better on the considered real-world datasets.
>
> If you are aware of another IVaR method that is applicable to the online/streaming setting that we have missed in our literature review, we would greatly appreciate if you could point it out specifically to us. We will be happy to compare against that in our revision as well. Thank you in advance.
>
> **W2** - First, we would like to point out that the identifiability conditions and moment assumptions we make are rather standard in the literature and can be satisfied in multiple cases (see, e.g., Lemma 1). It is also satisfied in our simulation experiments. Further relaxations of the moment and identifiability conditions have indeed been made in the literature in the offline settings. Taking your suggestion into account, we will add a discussion to them in our revision. A detailed study of developing algorithms and sample complexity results under relaxed assumptions is truly beyond the scope of this work.
>
> Regarding the robustness aspect, we point out under the setting in Section 4, identifiability conditions (Assumption 2.1) can be easily verified. Moment assumptions (Eq. (5), (6), (7), (8)) also hold:
> - (5)$$\begin{aligned}&\mathbb{E}\Big[\|X'X^\top-\mathbb{E}\_{X|Z}[X]\mathbb{E}\_{X|Z}[X]^\top\|^2\Big]\\\\
> =&\mathbb{E}\Big[\|c(h'+\epsilon_x')\phi(\gamma_{\ast}^\top Z)^\top+c\phi(\gamma_{\ast}^\top Z)(h+\epsilon_x)^\top+c^2(h'+\epsilon_x')(h+\epsilon_x)^\top-c\mathbf{1}\_{d_x}\phi(\gamma_{\ast}^\top Z)^\top-c\phi(\gamma_{\ast}^\top Z)\mathbf{1}\_{d_x}^\top-c^2\mathbf{1}\_{d_x}\mathbf{1}\_{d_x}^\top\|^2\Big]\\\\
> \leq&3\mathbb{E}\Big[\|c((h'+\epsilon_x')-\mathbf{1}\_{d_x})\phi(\gamma_{\ast}^\top Z)^\top\|^2\Big]+3\mathbb{E}\Big[\|c\phi(\gamma_{\ast}^\top Z)((h+\epsilon_x)-\mathbf{1}\_{d_x})^\top\|^2\Big]+3\mathbb{E}\Big[\|c^2((h'+\epsilon_x')(h+\epsilon_x)^\top-\mathbf{1}\_{d_x}\mathbf{1}\_{d_x}^\top)\|^2\Big]\\\\=&\mathcal{O}(c^2d^2+c^4d^2).
> \end{aligned}$$
> - (6)$$\begin{aligned}&\mathbb{E}\Big[\|YX'-\mathbb{E}\_{Y|Z}[Y]\mathbb{E}\_{X|Z}[X]\|^2\Big]\\\\=&\mathbb{E}\Big[\|(\theta_{\ast}^\top X+c\cdot(h_1+\epsilon_y))X'-(\theta_{\ast}^\top\mathbb{E}\_{X|Z}[X]+c)\mathbb{E}\_{X|Z}[X]\|^2\Big]\\\\\leq&3\mathbb{E}\Big[\|(X'X^\top-\mathbb{E}\_{X|Z}[X]\mathbb{E}\_{X|Z}[X]^\top)\theta_*\|^2\Big]+3c^2\mathbb{E}\Big[\|(h_1+\epsilon_y-1)\phi(\gamma_{\ast}^\top Z)\|^2\Big]+3c^4\mathbb{E}\Big[\|(h_1+\epsilon_y)(h'+\epsilon_x')-\mathbf{1}\_{d_x}\|^2\Big]\\\\=&\mathcal{O}(\|\theta_*\|^2(c^2d^2+c^4d^2)+c^2d_x^2+c^4).\end{aligned}$$
> - (7)$$\begin{aligned}&\mathbb{E}\Big[\|\mathbb{E}\_{X\mid Z}[X]\cdot\mathbb{E}\_{X\mid Z}[X]^\top-\mathbb{E}_Z\Big[\mathbb{E}\_{X\mid Z}[X]\cdot\mathbb{E}\_{X\mid Z}[X]^\top\Big]\|^2\Big]\\\\
> \leq&3\mathbb{E}\Big[\|\phi(\gamma\_{\ast}^\top Z)\phi(\gamma\_{\ast}^\top Z)^\top-\mathbb{E}\_{Z}\Big[\phi(\gamma\_{\ast}^\top Z)\phi(\gamma\_{\ast}^\top Z)^\top\Big]\|^2\Big]+3c^2\mathbb{E}\Big[\|\mathbf{1}\_{d_x}\phi(\gamma\_{\ast}^\top Z)^\top-\mathbb{E}\_Z\Big[\mathbf{1}\_{d_x}\phi(\gamma\_{\ast}^\top Z)^\top\Big]\|^2\Big]+3c^2\mathbb{E}\Big[\|\phi(\gamma\_{\ast}^\top Z)\mathbf{1}\_{d_x}^\top-\mathbb{E}\_{Z}\Big[\phi(\gamma\_{\ast}^\top Z)\mathbf{1}\_{d_x}^\top\Big]\|^2\Big]\\\\
> =&\mathcal{O}(d_z+c^2d_xd_z).\end{aligned}$$
> - (8)$$\begin{aligned}&\mathbb{E}\Big[\|\mathbb{E}\_{Y\mid Z}[Y]\cdot\mathbb{E}\_{X\mid Z}[X]-\mathbb{E}\_Z\Big[\mathbb{E}\_{Y\mid Z}[Y]\cdot\mathbb{E}\_{X\mid Z}[X]\Big]\|^2\Big]\\\\\leq&2\mathbb{E}\Big[\|(\mathbb{E}\_{X\mid Z}[X]\cdot\mathbb{E}\_{X\mid Z}[X]^\top-\mathbb{E}\_{Z}\Big[\mathbb{E}\_{X\mid Z}[X]\cdot\mathbb{E}\_{X\mid Z}[X]^\top\Big])\theta\_{\ast}\|^2\Big]+2c^2\mathbb{E}\Big[\|\phi(\gamma\_{\ast}^\top Z)-\mathbb{E}\_{Z}\Big[\phi(\gamma\_{\ast}^\top Z)\Big]\|^2\Big]\\\\=&\mathcal{O}((d_z+c^2d_xd_z)\|\theta_*\|^2+c^2d_z).
> \end{aligned}$$
>
> In particular, the constant $c$ represents the strength of endogeniety and as shown above we could precisely characterize how this strength of endogeniety affects the number of iterations (or samples in the streaming setting).  This demonstrates an instance of quantifying the robustness of the made assumptions (here the robustness is wrt to the constant $c \in (0,\infty)$. In the general response, we have also added a detailed discussion of Algorithm 2's performance under certain non-linear modeling assumptions, which is yet another robustness study of the proposed algorithms.
>
> ### **Questions**
>
> **Q1** - We would like to point out Proposition 1 extends the linear setting to the non-linear setting, in which we require boundedness of the variance of the gradient estimator $v(\theta)$. Relaxing the boundedness of the variance condition while still achieving convergence under the non-linear setting is challenging and left as our future work. Please also see the explanation under **Non-linear settings** in general response.
>
> We sincerely hope that our responses have answered your questions and concerns, in which case, we would appreciate if you could raise your scores appropriately. If you have additional questions, please reach out to us during the discussion period and we will be happy to clarify.

---

> > ### Comment · Reviewer_AcA7 · 2024-08-08
> >
> > Thank you very much for your detailed response. I will keep my original score.

---

### Official Review · Reviewer_WDCF · 2024-07-10

**Soundness:** 2
**Presentation:** 3
**Contribution:** 3
**Rating:** 6
**Confidence:** 3

**Summary:**

This paper tackles the instrumental variable regression. The problem setting assumes a model $Y=g_{\theta^*}(X) + \epsilon_1$, but unlike the ordinary regression model, there are correlations between $X$ and $\epsilon_1$. The model assumes in addition an instrument variable $Z$ such that $Y$ and $X$ are independent conditional on $Z$, and $X=h_{\gamma^*}(Z)+\epsilon_2$. The target is to estimate $\theta^*$. The canonical approach is to use the two-stage least square (2SLS) method, where we first regress $X$ on $Z$ and then regress $Y$ on the $\widehat{X}$ that is estimated by $Z$.

This paper considers an online setting where samples $(X_t, Y_t, Z_t)$ arrive sequentially. They propose two stochastic gradient descent (SGD) based algorithms. The first algorithm assumes that for each $Z_t$ we can resample $X_t'$ from the conditional distribution of $X | Z=Z_t$. The second algorithm does not make this assumption and can be considered an SGD version of the 2SLS algorithm. The paper provides the L2 convergence guarantee for both algorithms and they design experiments to demonstrate the effectiveness of their algorithms.

**Strengths:**

This paper is clearly written and contains rigorous theoretical analysis. It presents online algorithms to address the instrumental variable regression problem. Compared to their offline counterparts, these online algorithms are more memory-efficient and computationally stable. The proof of their Theorem 2 introduces an intermediate sequence $\widetilde{\theta}_t$ with known dynamic, and evaluates the convergence rate of $|\theta_t - \widetilde{\theta}_t|$. This technique is of separate interest.

**Weaknesses:**

1.  The contribution of the paper appears marginal. The first algorithm requires data resampling and this limits its applicability. The second algorithm is an intuitive adaption of the 2SLS algorithm to the SGD setting. The theorems are proved in the simple linear setting.
2. Apart from avoiding matrix inversion, the paper lacks necessary explanations as to why we should prefer an SGD version of 2SLS over the canonical offline 2SLS. It would be beneficial if more content were devoted to the advantages of Algorithm 2 compared to traditional benchmarks. For example, a theoretical comparison with 2SLS or an empirical comparison using real-world data would be helpful.
3. Could you explain why using two different synthetic settings for Algorithm 1 and Algorithm 2 in the Numerical Experiments section?

**Questions:**

See "Weakness".

**Limitations:**

See "Weakness".

---

> ### Author Rebuttal · Authors · 2024-08-07
>
> Dear Reviewer WDCF,
>
> Thank you for your review. We provide responses to your concerns in 'weaknesses'.
>
> **W1** - We strongly disagree with your view that the paper's contribution is marginal. As noted in Prop 1, Alg. 1 is not restricted to linear models; it can handle non-linear and non-convex cases, including DNNs, without explicitly specifying or estimating the model between $Z$ and $X$.
>
> We also contest your claim that Algorithm 1's data resampling limits its applicability. Instead, it offers a novel approach for online data collection to avoid forbidden regression issues.
>
> Regarding the proof techniques for Alg 1 (Thm 1), classical SGD convergence methods do not apply directly because the gradient estimator $v(\theta)$ does not meet the bounded variance  ([Lan20])  or expected smoothness ([KR20]) assumptions. We provide a novel analysis based on weaker statistical assumptions for streaming data compared to those in the optimization literature.
>
> We now highlight the additional challenges in the proof of Alg. 2 (see, also, the part after Remark 1).
>
> **Challenge 1 (Interaction between iterates):** The major challenge towards the  convergence analysis of $\\{\theta_t\\}_t$ lies in the interaction term $\gamma_t Z_tZ_t^\top\gamma_t\theta_t$ between $\gamma_t$ and $\theta_t$ in equation (13). Note that this multiplicative interaction term is neither a martingale-difference sequence not does it have finite variance which could have led to a simpler analysis. This involved dependence between the noise in the stochastic gradient updates for the two stages does not appear in existing problem setups and corresponding analysis of non-linear two time-scale algorithms [MP06,DTSM18,Doa22] (more related references in the paper).
>
> **Challenge 2 (Biased Stochastic Gradient):**
> In our setting, as shown below, the stochastic gradient in equation (13) evaluated at $(\theta_t,\gamma_t)$ is biased unlike existing works on two-time-scale algorithms (see [MP06,DTSM18,Doa22] for example).
>
> $$
> \begin{aligned}
> &\mathbb{E}\_{t,Z_t}[\gamma_t^\top Z_t(Z_t^\top\gamma_t\theta_t-Y_t)]=\mathbb{E}\_{t,Z_t}[\gamma_t^\top Z_t(Z_t^\top\gamma_t\theta_t-Z_t^\top\gamma_*\theta_*)]=\mathbb{E}\_{t}[\gamma_t^\top\Sigma_Z(\gamma_t\theta_t-\gamma_*\theta_*)]\\\\=&\gamma_t^\top\Sigma_Z\gamma_t(\theta_t-\theta_*)+\gamma_t^\top\Sigma_Z(\gamma_t-\gamma_*)\theta_* \neq \gamma_{\ast}^\top\Sigma_Z\gamma_*(\theta_t-\theta_*) =\nabla_\theta F(\theta_t).
> \end{aligned}
> $$
>
> **Challenge 3 (No Bounded Variance Assumption of Stochastic Gradient):**
> We do not assume boundedness of $\{\theta_t\}$ iterates unlike existing works (see Assumption 1 in [WZZ21], Assumption 2 in [XL21] and Theorem 2 in [MSBPSS09]). This assumption ensures uniform boundedness of the variance of the stochastic gradient requiring a simpler analysis compared to us.
>
> Resolving these issues firstly require proving that $\mathbb{E}[\|\theta_t\|_2^4]$ is bounded (Lemma 5) which is non-trivial and requires carefully chosen stepsizes satisfying
>
> $\sum_{t=1}^\infty(\alpha_t^2+\alpha_t\sqrt{\beta_t})<\infty.$
>
> Using this bound on $\mathbb{E}[\|\theta_t\|_2^4]$, we prove the convergence of sequence $\delta_t\coloneqq \theta_t-\tilde{\theta}_t$, the error between true iterates and auxiliary iterates we defined. This requires a novel proof technique where we first provide an intermediate bound (see Lemma 6)  and then progressively sharpen to a tighter bound (see Lemma 7).
>
> [MSBPSS09] H. Maei, C. Szepesvari, S. Bhatnagar, D. Precup, D. Silver, and R. Sutton. Convergent temporal-difference learning with arbitrary smooth function approximation. NeurIPS 2009
>
> [Lan20] G. Lan. First-order and stochastic optimization methods for machine learning. Springer, 2020
>
> [KR20] A. Khaled and P. Richtárik. Better theory for SGD in the nonconvex world.
>
> **W2** - Alg. 2, the SGD version of 2SLS is applicable for various emerging applications of online/streaming IVaR, i.e., mobile health applications like Just-In-Time Adaptive Interventions (e.g., [TM17],[DVB23a]). Comparing to other online IVaR methods, ours achieve a much better per-iteration computational and memory costs as illustrated in Appendix B Table 1.
>
> From an algorithmic perspective, integrating non-linear models into offline 2SLS requires additional optimization, typically handled by Stochastic Gradient Descent (SGD) methods. Recent research in both machine learning (e.g., [VSH+16, DVB23a]) and economics (e.g., [CLL+23]) has focused on developing online methods for IVaR.  In our general response, we outlined the specific non-linear models which could be handled by our Alg. 2. Note that Alg. 1 already handles non-linear models (see Prop 1). We believe our work significantly advances streaming IV regression and expect further research to extend our methodology to more general non-linear settings.
>
> In addition, we added experiments of Alg 2 on real-world dataset. Please see the general response and the attached PDF for the results.
>
> **W3** - Alg. 1 and Alg. 2 have different settings (oracles, assumptions, etc.). In Alg. 1, we assume that we have access to 2-sample oracles (i.e., 2 independent samples $X, X'$ can be drawn given $Z$), and the algorithm can handle the cases when $g(\theta; X)$ is linear (Thm 1) and non-linear (Prop 1). Hence we consider both linear ($\phi(s) = s$) and non-linear ($\phi(s) = s^2$) settings in the experiments of Alg. 1. In Alg. 2, we mainly consider the case when we only have access to 1-sample oracle (i.e., 1 sample $X$ can be drawn given $Z$), and the model is linear.
>
> [TM17] A. Tewari, and S. A. Murphy. "From ads to interventions: Contextual bandits in mobile health." Mobile health: sensors, analytic methods, and applications (2017)
>
> We sincerely hope that our responses have answered your questions and concerns, in which case, we would appreciate if you could raise your scores appropriately. If you have additional questions, please reach out to us during the discussion period and we will be happy to clarify.

---

> > ### Comment · Reviewer_WDCF · 2024-08-10
> >
> > Thank you for the detailed reply, I am now more convinced of the paper's technical contribution. I will raise my score from 5 to 6.

---

### Official Review · Reviewer_hwGZ · 2024-07-11

**Soundness:** 3
**Presentation:** 3
**Contribution:** 3
**Rating:** 5
**Confidence:** 1

**Summary:**

The paper shows algorithms for instrumental variable regression that dont need matirx inversions and mini-batches. At the same time, the paper give the rates of convergence.

**Strengths:**

The proposed method offers robust theoretical guarantees and is validated through comprehensive experimental results.

**Weaknesses:**

None

**Questions:**

I am curious about how the matrix inversion harm a algorithm performance in this setting.

---

> ### Author Rebuttal · Authors · 2024-08-07
>
> Dear Reviewer hwGZ,
>
> Thank you for your comments. Below we provide our response to your questions.
>
> ### **Question**
>
> >Q1. Matrix inversion
>
> **Response:**
>
> Please kindly refer to Appendix B for a summary table (Table 1) on arithmetic operations complexity per-iteration. Using the method with matrix inversion (DVB23b, v1), it requires $d_x^3+t d_x^2$ arithmetic operations where $d_x$ is the dimension of the input $X$ and $t$ denotes the $t$-th iteration. Using the method in their updated version (DVB23a, v3), the computational and memory complexity per-iteration is still much higher than our methods. This quantifies the inefficiency of matrix-inversion based methods.
>
> We have also provided real-world experiments demonstrating the improved and efficient performance of our algorithm over other approaches.
>
> We sincerely hope that our responses have answered your questions and concerns, in which case, we would appreciate if you could raise your scores appropriately. If you have additional questions, please reach out to us during the discussion period and we will be happy to clarify.

---

### Official Review · Reviewer_E97U · 2024-07-12

**Soundness:** 3
**Presentation:** 3
**Contribution:** 3
**Rating:** 6
**Confidence:** 3

**Summary:**

This paper proposes and analyzes on-line algorithms for instrumental variable regression (IVaR) with streaming data.  Specifically, the authors consider the model:

$Y = g_{\theta^*}(X) + \epsilon_1$

where the covariate $X$ and noise $\epsilon_1$ are possibly correlated, but an instrumental variable $Z$ is available that satisfies:

$X = h_{\gamma^*}(Z) + \epsilon_2$

with $\epsilon_2$ being a centered (unobserved) noise.  Building on the prior works [MMLR20] and [HZCH20], the authors consider the IVaR problem formulated as a conditional stochastic optimization problem, presented in Equation (2):

$Minimize_g F(g) := E_Z E_{Y|Z} [ ( Y - E_{X|Z} [g(X)])^2 ].$

Considering a parameterized family of regression functions $G := \{ g(\theta; X) | \theta \}$, the authors observe that the gradient of F admits the expression as in Equation (3):

$\nabla F(\theta) = E_Z[ (E_{X|Z}[g(\theta;X)] - E_{Y|Z}[Y]) \cdot \nabla_{\theta} E_{X|Z}[g(\theta; X)]].$


This paper propose and analyze two streaming algorithms to solve the optimization problem formulation above. Notably, the proposals in this work do not require reformulating it to a minimax optimization problem as done in [MMLR20] or employing a nested sampling technique to reduce the bias in gradient estimation as in [HZCH20]. Specifically, the authors assume the availability of an oracle that can generate a sample $X$ (or two independent samples $X$ and $X’$) conditioned on $Z$, and then propose a stochastic gradient descent for IVaR. Additionally, the authors establish the rate of convergence assuming linear models. The proposed algorithms and claims are supported by numerical experiments with synthetic data.

**Strengths:**

This work provides a simple yet effective algorithmic solution to solve the IVaR problem formulated as a stochastic optimization problem, overcoming challenges highlighted in prior work [MMLR20] and adapting the method in [HZCH20] for streaming settings. This avoidance of the need for nested sampling (=generating batches) is enabled by leveraging the structure of the quadratic loss in the gradient expression.

The paper is well-organized and presents its core ideas clearly. Section 2 introduces the two-sample oracle assumption, which, while somewhat idealistic, is reasonable for discrete-valued $Z$ as remarked by the authors. Section 3 then transitions to a more realistic one-sample oracle, focusing on linear models and modifying the algorithm and analysis from Section 2 accordingly. All required assumptions and propositions are stated explicitly and clearly.

**Weaknesses:**

While this work makes several significant theoretical contributions, especially in advancing the analysis, there are areas for potential improvement:

**1. Motivation for IVaR Problems (with Streaming Data)**: The importance of IVaR problems, especially with streaming data, should be highlighted more. Discussing example scenarios in addition to citing references would better motivate and convince readers of the problem's relevance.

**2. Further Experimental Validation**: Although this work is primarily theoretical, augmenting the experiment section with a more comprehensive set of experiments would be beneficial. It would be particularly valuable to see how the proposed algorithms perform on real-world datasets.

**Questions:**

1. While Algorithm 1 is generally applicable with the availability of two-sample oracles, Algorithm 2 seems to hinge critically on linear models. The authors note in lines 244-245 that a detailed treatment of the nonlinear case is left for future work due to its complexity in analysis, but I am curious if designing a working algorithm based on the insights in this paper would be feasible at least. Could the authors provide insights on the following:

    (a) How would Algorithm 2 (or a variant) perform with nonlinear models?

    (b) Do the authors believe extending Algorithm 2 to nonlinear models is feasible, and what challenges might arise in the algorithm's design?

2. In line 57, can authors elaborate on why "Eq. (3) implies that one does not need the nested sampling technique to reduce the bias" with more details?

3. Suggestions

    (a) **Lines 150 - 151**:  I suggest the authors state Assumption 2.3 as follows: "... $P_{Y|X}$.  There exist constants such that $C_x, C_y, C_{xx}, C_{yx} > 0$ such that ..."

    (b) **Line 161**:  I guess it might make more sense to move Assumption 2.4 up to Line 147, right after the sentence "... pre-collected dataset."

    (c) **Page 9**: The authors may want to summarize the information in Lines 315 - 323 in the captions of Figures 3 and 4 for readers' convenience.  Also, it could be helpful to summarize the experimental results in the main text explicitly.

    (d) **Miscellaneous/Potential typos**:

        i) Line 52: $X|Z$ instead of $Z|X$?

        ii) Line 154: remove "under"

        iii) Eq. (10) and Line 159: Please consider ending the sentence in Eq. (1) and start a new sentence in Line 159. Also, "if" in Line 159 should be capitalized.

**Limitations:**

This is primarily a theoretical work, and the authors discussed the potential limitations of their assumptions.

---

> ### Author Rebuttal · Authors · 2024-08-07
>
> Dear Reviewer E97U,
>
> Thank you for your insightful review. Below we provide our response to your concerns and questions.
>
> ### **Weaknesses**
>
> >W1. Motivation
>
> **Response**:
>
> As mentioned in the general response "Adv 3 - Emerging applications", a motivation for developing online/streaming IVaR is that of mobile health applications like Just-In-Time Adaptive Interventions (see, e.g., [TM17]) which also serves as a motivation for the work of [DVB23a]. Our work is directly applicable for several such applications. Taking your suggestion into account, we will be adding a discussion about this application in detail in our revision.
>
> [TM17] Tewari, Ambuj, and Susan A. Murphy. "From ads to interventions: Contextual bandits in mobile health." Mobile health: sensors, analytic methods, and applications (2017): 495-517.
>
>
> >W2. Experimental Validation
>
> **Response**:
>
> Please see our general response where we have added two real-data experiments. The results show the benefits of the proposed algorithms, thereby supporting the theoretical results.
>
> ### **Questions**
>
> > Q1. While Algorithm 1 is...
>
> **Response**:
>
> We thank the review for asking this insightful question. Please see the explanation under **Non-linear settings** in the general response. We will add the above explanation as a remark in our revision.
>
> >Q2. In line 57...
>
> **Response**:
>
> For general conditional stochastic optimization, its objective and gradient are of the form
>
> $$F(\theta) = \mathbb{E}_{Z} l (\mathbb{E}\_{X\mid Z} [g(\theta;X)] - \mathbb{E}\_{Y \mid Z} [Y])$$
>
> and
>
> $$\nabla F(\theta) = \mathbb{E}_{Z}[\nabla\_{\theta} \mathbb{E}\_{X\mid Z}[g(\theta;X)] \nabla\_{\theta} l(\mathbb{E}\_{X\mid Z}[g(\theta;X)] - \mathbb{E}\_{Y\mid Z} [Y])],$$
>
> where $l$ is a non-linear and non-quadratic function. Therefore, $\nabla_\theta l$ is non-linear. For the composition of non-linear function and expectation term $\nabla\_\theta l(\mathbb{E}\_{X\mid Z}[g(\theta;X)] - \mathbb{E}\_{Y\mid Z} [Y])$ for a given $Z$, getting a low-bias estimator requires a large batch of samples from the conditional distribution of $(X,Y)$ given $Z$ for the following observation: $\|\nabla\_\theta l(\mathbb{E}\_{X\mid Z}[g(\theta;X)] - \mathbb{E}\_{Y\mid Z} [Y]) - \mathbb{E}\nabla\_\theta l(\frac{1}{m}\sum_{i=1}^m g(\theta;X_i)- \frac{1}{m}\sum_{i=1}^m Y_i)\|$ $\leq S_l \mathbb{E}\|\mathbb{E}\_{X\mid Z}[g(\theta;X)]  -\frac{1}{m}\sum_{i=1}^m g(\theta;X_i) + \frac{1}{m}\sum_{i=1}^m Y_i- \mathbb{E}\_{Y\mid Z} [Y]\|=\mathcal{O}(1/\sqrt{m})$. Here $S_l$ denotes the Lipschitz smooth parameter. However, when $l$ is just $\|\cdot\|^2$ as in the instrumental variable regression, $\nabla l$ is linear. Thus the right-hand-side of the above inequality simplifies to $0$, meaning that for any $m$, it is an unbiased estimator. Thus it suffices to use $m=1$ and avoids to use batch.  This observation, while being straightforward in hindsight,  has not been observed in previous work (which all hence used more complicated algorithmic design).
>
>
> >Q3. Suggestions
>
> **Response**:
>
> We really appreciate your careful reading and insightful suggestions. We will incorporate them in our revision.
>
> We sincerely hope that our responses have answered your questions and concerns, in which case, we would appreciate if you could raise your scores appropriately. If you have additional questions, please reach out to us during the discussion period and we will be happy to clarify.

---

> > ### Comment · Reviewer_E97U · 2024-08-10
> > **Response to the Authors' Rebuttal**
> >
> > I thank the authors for addressing my concerns and questions.  I find most of their responses satisfactory, and would like to encourage them to incorporate the explanations and remarks into the revision.
> >
> > However, I am still not clearly seeing how Algorithm 2 can be immediately applied to the scenario where the relationship between $X$ and $Z$ is non-linear, as the authors claim.  Specifically, the update rules in Eqs. (13) and (14) -- which stem from Eq. (11) and include an additional modification trick to promote stability as discussed in Lines 220 -- 226 -- seem to rely on the linear model assumption.  Thus, it is unclear to me how these would translate to the non-linear model setting described by the authors, namely, the setting where $Y = {\theta^*}^{\top} X + \epsilon_1$ with $X = h_{\gamma^*}(X) + \epsilon_2$.  I believe the update rule should involve $\nabla h_{\gamma_t}$ and possibly a similar modification to promote algorithmic stability.
> >
> > Could you please clarify the update rules for this setting by specifying the counterparts of Eqs. (13) and (14), and any necessary modifications (if needed) that corresponds to replacing $g(\theta_t, X_t) = X_t^{\top} \theta_t$ with $Z_t^{\top} \gamma_t \theta_t$ in the linear setting?

---

> > > ### Author Response · Authors · 2024-08-12
> > > **Thank you for the insightful question**
> > >
> > > It is indeed true that learning  $\gamma_*$ requires incorporating $\nabla h$ in the update equation as we have alluded to in our global response in the section "**Alg. 2 convergence proof intuition**" but could not elaborate on due to lack of space. We elaborate on it here.
> > >
> > > **The same stability issue happens in this framework as we saw in the linear setting (Line 220 - 226) and is avoided by the same trick that we used in the linear setting.**
> > > Consider the following version of the equation (11) adapted to the non-linear setting.
> > >  $$\theta_{t+1}=\theta_t-\alpha_{t+1} h_{\gamma_t}(Z_t)(X_t^\top \theta_t-Y_t)\qquad
> > >  \gamma_{t+1}=\gamma_t-\beta_{t+1} \nabla h_{\gamma_t}(Z_t)^\top( h_{\gamma_t}(Z_t)-X_t).\qquad \text{(11NL)}$$
> > > Equation (11NL), similar to equation (12) in the main paper,  can be expanded in the following manner.
> > >
> > > $$\theta_{t+1}-\theta_* = {\hat{Q}}\_t^{NL} ( \theta_t-\theta_* )+\alpha_{t+1}\mathbb{E}\_{\gamma_t}[(h_{\gamma_t}(Z_t)-h_{\gamma_*}(Z_t)) Y_t)]+\alpha_{t+1}D_t^{NL}\theta_*+\alpha_{t+1}\left(\mathbb{E}\_{\gamma_t}[h_{\gamma_t}(Z_t)h_{\gamma_*}(Z_t)^\top]-h_{\gamma_t}(Z_t)h_{\gamma_*}(Z_t)^\top\right)(\theta_t-\theta_*)$$ $$~~~~~~~~~~~~~~~~~~~~+\alpha_{t+1}\left(\mathbb{E}\_{\gamma_t}[h_{\gamma_t}(Z_t)h_{\gamma_*}(Z_t)^\top]-h_{\gamma_t}(Z_t)h_{\gamma_*}(Z_t)^\top\right)\theta_*+\alpha_{t+1}((h_{\gamma_t}(Z_t)-\mathbb{E}\_{\gamma_t}[h_{\gamma_t}(Z_t)])Y_t)
> > > -\alpha_{t+1}h_{\gamma_t}(Z_t)\epsilon_{2,t}^\top\theta_t. \qquad \text{(12NL)}$$
> > >
> > > where $\hat{Q}\_t^{NL}=(I-\alpha_{t+1}\mathbb{E}\_{\gamma_t}[h_{\gamma_t}(Z_t)h_{\gamma_*}(Z_t)^\top])$, $D_t^{NL}=\mathbb{E}\_{\gamma_t}[(h_{\gamma_*}(Z_t)-h_{\gamma_t}(Z_t)))h_{\gamma_*}(Z_t)^\top]$.
> > >
> > > First, let's focus on the **stability issue** associated with the first term on the RHS, i.e., $\hat{Q}\_t^{NL}(\theta_t-\theta_*)$. Just like the linear setting, here too, the matrix $\mathbb{E}\_{\gamma_t}[h_{\gamma_t}(Z_t)h_{\gamma_*}(Z_t)^\top]$ is not guaranteed to be positive semi-definite. So, we replace the term $X_t^\top\theta_t$ in equation (11NL) by $h_{\gamma_t}(Z_t)^\top\theta_t$ which leads to the following modified Algorithm 2 updates.
> > > $$\theta_{t+1}=\theta_t-\alpha_{t+1} h_{\gamma_t}(Z_t)(h_{\gamma_t}(Z_t)^\top \theta_t-Y_t)\qquad \text{(13NL)}$$
> > > $$\gamma_{t+1}=\gamma_t-\beta_{t+1} \nabla h_{\gamma_t}(Z_t)^\top( h_{\gamma_t}(Z_t)-X_t),\qquad \text{(14NL)}$$
> > > where $\gamma_t\in\mathbb{R}^{d_\gamma}$, and $\nabla h_{\gamma_t}(Z_t)\in\mathbb{R}^{d_x\times d_\gamma}$ is the Jacobian of $h$ with respect to $\gamma_t$.
> > >
> > > For (13NL), we will have $\hat{Q}\_t^{NL}$ of the form $\hat{Q}\_t^{NL}=(I-\alpha_{t+1}\mathbb{E}\_{\gamma_t}[h_{\gamma_t}(Z_t)h_{\gamma_t}(Z_t)^\top])$. Here $\mathbb{E}\_{\gamma_t}[h_{\gamma_t}(Z_t)h_{\gamma_t}(Z_t)^\top]$ is positive semi-definite leading to the stability of the dynamics just like the linear case.
> > >
> > > **Now, we just have to show that rest of the terms on the right hand side of (12NL) converges similar to (12).** Recall that, except the first term, we control all the other terms on the right hand side of equation (12) mainly by using the martingale-difference property of the noise variables, and Lemma 3, i.e. the convergence of $\mathbb{E}[\||\gamma_t-\gamma_*\||^2]$. In (12NL), the martingale-difference property of the noise variables in the fourth to seventh terms on the right hand side clearly holds.
> > >
> > > It remains to obtain a result analogous to Lemma 3. To do so, we look at equation (14NL). Analysis of equation (14NL) to establish a convergence rate for the $\gamma_t$ updates to $\gamma_*$ or $h_{\gamma_t}$ to $h_{\gamma_*}$ is straightforward as long as $H(\gamma)\coloneqq\mathbb{E}[\||X-h_{\gamma}(Z)\||^2]$ is strongly-convex [PJ92], or satisfies Polyak-Łojasiewicz (PL) inequality [KNS16]. This increases the model flexibility considerably as PL inequalities are satisfied for a wide class of non-linear DNN models [LZB22]. Beyond strong-convex and PL cases, the analysis is challenging although the same algorithmic framework still applies as a methodology.
> > >
> > > Putting the above pieces together, it is possible to obtain the rates of convergence for the case when $Z$ to $X$ is non-linear.
> > >
> > > Please reach out to us if you have any additional question.

---

> > > > ### Author Response · Authors · 2024-08-13
> > > >
> > > > Dear Reviewer,
> > > >
> > > > Since the discussion period is about to end in a day, we wanted to make sure that our response fully addresses your question. If we have answered your question satisfactorily, we would appreciate if you consider increasing your score as you see fit.
> > > >
> > > > Thank you.

---

### Author Rebuttal · Authors · 2024-08-07

Dear reviewers,

Thank you for your comments and questions. Below we provide our general response. We first present real-data experiments, and then re-emphasize points which were potentially overlooked.

### **Real Data Examples**
We illustrate Alg. 2 on 2 datasets: Angrist and Evans (1998) Children/Parents' Labor Supply Data and U.S.Portland Cement Industry Data. Please see the global response PDF (attached) for the plots and experimental details.
### **Advantages of our approach**
 - Avoiding "forbidden regression": Note that one of the main benefits of the online approach to IVaR that we discovered in our work is that of avoiding "forbidden regression" under the 2-sample oracle. To our knowledge, this solution is not available for any existing online and offline procedures in the current literature, and provides a novel data-collection mechanism for practitioners.

 - Computational benefits: SGD-type algorithms have played a crucial role in scaling up statistical ML methods. Recent works ML (e.g., [VSH+16, DVB23a]) and economics (e.g., [CLL+23]) develop online IVaR methods. Our algorithms have state-of-the-art computational complexities (Appendix B) for online IVaR in linear settings, even compared to offline procedures like 2SLS.

 - Emerging applications: Another motivation for developing streaming IVaR is mobile health applications like Just-In-Time Adaptive Interventions (see, [TM17] and [DVB23]). Our work is directly applicable for several such applications. We will add a discussion about these applications in our revision.

[TM17] A. Tewari, and S. A. Murphy. "From ads to interventions: Contextual bandits in mobile health." Mobile health: sensors, analytic methods, and applications (2017): 495-517.

### **Non-linear settings**

Our work is not restricted to just the linear setting. For Alg. 1, Prop. 1 extends the results to the non-linear setting.

**For Alg. 2, it can be immediately applied to the setting where the relation between $X$ and $Z$ is non-linear.** Consider the model:
\begin{align*}
Y=\theta_{\ast}^\top X+\epsilon_1\quad\text{with}\quad X=h_{\gamma_*}(Z)+\epsilon_2
\end{align*}

### **Unbiasedness of the Alg. 2 output**

Assume that one can learn $h_{\gamma_*}$ efficiently, i.e., say we have $h_{\gamma_t}\approx h_{\gamma_*}$ for some large $t$ (see below for the required conditions). Considering the update for $\theta_t$ in Algorithm 2 (Eq. (13)), we have
$$\theta_{t+1}=\theta_t-\alpha_{t+1} h_{\gamma_t}(Z_t)(h_{\gamma_t}(Z_t)^\top \theta_t-Y_t)\approx \theta_t-\alpha_{t+1} h_{\gamma_*}(Z_t)(h_{\gamma_*}(Z_t)^\top \theta_t-Y_t).$$
It is easy to see that the limit point $\theta_\infty$ of this update will satisfy, $$\mathbb{E}[h_{\gamma_*}(Z)h_{\gamma_*}(Z)^\top]\theta_\infty= \mathbb{E}[h_{\gamma_*}(Z)h_{\gamma_*}(Z)^\top]\theta_*$$
which implies that $\theta_t$ is an unbiased estimator of $\theta_*$ as long as $\mathbb{E}[h_{\gamma_*}(Z)h_{\gamma_*}(Z)^\top]$ is invertible (Assumption 2.2).

### **Alg. 2 convergence proof intuition**
In the proof of Thm. 2, the major challenge is to control the interaction term $\gamma_t Z_tZ_t^\top\gamma_t\theta_t$. We use Lemma 3 which establishes the convergence rate of $\mathbb{E}[\|\gamma_t-\gamma_*\|^2]$. Analogously, in the nonlinear setting, we need to prove the convergence of $\gamma_t$ to $\gamma_*$ or $h_{\gamma_t}(Z)$ to $h_{\gamma_*}(Z)$. This is possible when the loss function $\mathbb{E}[\|X-h_{\gamma}(Z)\|^2]$ is strongly-convex, or satisfies Polyak-Łojasiewicz (PL) inequality [KNS16]. This allows for considerable flexibility in the model choices including basis expansion based non-linear methods and wide Neural Networks [LZB22]. Since the update equation for $\gamma_t$ (Eq (14)) does not involve $\theta_t$, the rest of the proof follows by similar techniques except for the obvious changes required due to the above modification, e.g., the term $\gamma_{\ast}^\top\Sigma_Z\gamma_*$ will be replaced by $\mathbb{E}[h_{\gamma_*}(Z)h_{\gamma_*}(Z)^\top]$.


[KNS16] H. Karimi, J. Nutini, and M. Schmidt. "Linear convergence of gradient and proximal-gradient methods under the polyak-łojasiewicz condition." In ECML PKDD 2016.

[LZB22] C. Liu, L. Zhu, and M. Belkin. "Loss landscapes and optimization in over-parameterized non-linear systems and neural networks." Applied and Computational Harmonic Analysis 59 (2022): 85-116.

Now, consider the case where the upper level is non-linear, i.e.,
\begin{align*}
    Y=g_{\theta_*}(X)+\epsilon_1\quad\text{with}\quad X=\gamma_{\ast}^\top Z+\epsilon_2.
\end{align*}
In this case, the lower level problem can be solved efficiently as it is a simple linear regression problem. For brevity, assume that $\gamma_*$ is known. In that case, the update for $\theta_t$ in Alg. 2 takes the following form.
\begin{align*}
    \theta_{t+1}=\theta_t-\alpha_{t+1}\nabla g_\theta(Z_t^\top\gamma_*)(g_{\theta}(Z_t^\top\gamma_*)-Y_t).
\end{align*}
Under suitable convergence conditions, this update finds the root of $$\mathbb{E}[\nabla g_\theta(Z^\top\gamma_*)(g_\theta(Z^\top\gamma_*)-Y)]=0,$$ or equivalently, the root of,
\begin{align*}
    &\mathbb{E}[\nabla g_\theta(Z^\top\gamma_*)(g_\theta(Z^\top\gamma_*)-g_{\theta_*}(Z^\top\gamma_*+\epsilon_2)-\epsilon_1)]=0\\
    &\mathbb{E}[\nabla g_\theta(Z^\top\gamma_*)g_\theta(Z^\top\gamma_*)]=\mathbb{E}[\nabla g_\theta(Z^\top\gamma_*)g_{\theta_*}(Z^\top\gamma_*+\epsilon_2)].
\end{align*}
It is easy to see that for general non-linear $g$, $\theta=\theta_*$ may not be a solution to the above equation, i.e., Alg. 2 will find a biased solution. This demands a modified version of Alg. 2 involving an additional debiasing step. As the analysis of Alg. 2 is already quite challenging for the model where the upper layer is linear (see "Proof Techniques" after Remark 1 in the paper and our reply to Reviewer WDCF), we defer the analysis of the case where $Y$ depends non-linearly on $X$ to the future work.

---

### Decision · Program_Chairs · 2024-09-25

**Decision:**

Accept (poster)

**Comment:**

Four reviewers have reviewed the paper, and their overall evaluation is positive. I concur with their assessment and believe the paper makes a strong contribution with compelling results. Two reviewers requested additional experiments and comparisons, to which the authors have responded satisfactorily. I strongly recommend that the authors include these additional simulations in the final version of the paper.